# SIESTA: Efficient Online Continual Learning with Sleep

**Md Yousuf Harun** *                                                   *mh1023@rit.edu*
*Rochester Institute of Technology*

**Jhair Gallardo** *                                                    *gg4099@rit.edu*
*Rochester Institute of Technology*

**Tyler L. Hayes** †                                                    *tlh6792@rit.edu*
*Rochester Institute of Technology*

**Ronald Kemker**                                           *ronald.kemker@spaceforce.mil*
*United States Space Force*

**Christopher Kanan**                                         *ckanan@cs.rochester.edu*
*University of Rochester*

**Reviewed on OpenReview:** *https://openreview.net/forum?id=MqDVlBWRRV*

## Abstract

In supervised continual learning, a deep neural network (DNN) is updated with an ever-growing data stream. Unlike the offline setting where data is shuffled, we cannot make any distributional assumptions about the data stream. Ideally, only one pass through the dataset is needed for computational efficiency. However, existing methods are inadequate and make many assumptions that cannot be made for real-world applications, while simultaneously failing to improve computational efficiency. In this paper, we propose a novel continual learning method, SIESTA based on wake/sleep framework for training, which is well aligned to the needs of on-device learning. The major goal of SIESTA is to advance compute efficient continual learning so that DNNs can be updated efficiently using far less time and energy. The principal innovations of SIESTA are: 1) rapid online updates using a rehearsal-free, backpropagation-free, and data-driven network update rule during its wake phase, and 2) expedited memory consolidation using a compute-restricted rehearsal policy during its sleep phase. For memory efficiency, SIESTA adapts latent rehearsal using memory indexing from REMIND. Compared to REMIND and prior arts, SIESTA is far more computationally efficient, enabling continual learning on ImageNet-1K in under 2 hours on a single GPU; moreover, in the augmentation-free setting it matches the performance of the offline learner, a milestone critical to driving adoption of continual learning in real-world applications [1].

## 1 Introduction

Training DNNs is incredibly resource intensive. This is true for both learning in highly resource constrained settings, e.g., on-device learning, and for training large production-level DNNs that can require weeks of expensive cloud compute. Moreover, for real-world applications, the amount of training data typically grows over time. This is often tackled by periodically re-training these production systems from scratch, which requires ever-growing computational resources as the dataset increases in size. Continual learning algorithms have the ability to learn from ever-growing data streams, and they have been argued as a potential solution for efficient learning for both embedded and large production-level DNN systems, improving the computational efficiency of network training and updating (Parisi et al., 2019). However, continual learning is rarely used for real-world applications because these algorithms fail to achieve

---

*Equal contribution.

†Now at NAVER LABS Europe.

[1]Code is available at `https://yousuf907.github.io/siestasite`

comparable performance to offline retraining or they make assumptions that do not match real-world applications. As shown in Harun et al. (2023a), many state-of-the-art continual learning methods e.g., BiC (Wu et al., 2019), WA (Zhao et al., 2020), and DER (Yan et al., 2021) are more expensive than offline models trained from scratch. Recent works have also argued that compute needs to be the focus of continual learning and that constraining memory serves little purpose except for on-device learning because storage costs are negligible compared to computation when training DNNs (Prabhu et al., 2023a;b; Hammoud et al., 2023). In this paper, we describe a resource efficient, continual learning algorithm that rivals an offline learner on supervised tasks, a critical milestone toward enabling the use of continual learning for real-world applications.

In conventional offline training of DNNs, training data is shuffled (making it independent and identically distributed (iid), which is required for stochastic gradient descent (SGD) optimization) and repeatedly looped through many training iterations. In contrast, an ideal continual learning algorithm is able to efficiently learn from *potentially* non-iid data streams, where each training sample is only seen by the learner once unless a limited amount of auxiliary storage is used to cache it. Most continual learning algorithms are designed solely to overcome catastrophic forgetting, which occurs when training with non-iid data (Parisi et al., 2019; Kemker et al., 2018). To do this, most models make implicit or explicit assumptions that go beyond the general supervised learning setting, e.g., some methods assume the availability of additional information or assume a specific structure of the data stream. Moreover, existing methods do not match the performance of an offline learner, which is essential for industry to adopt continual learning for updating large DNNs.

We argue that a continual learning algorithm should have the following properties:

1. It should be capable of online learning and inference in a compute and memory constrained environment,
2. It should rival (or exceed) an offline learner, regardless of the structure of the training data stream,
3. It should be significantly more computationally efficient than training from scratch, and
4. It should make no additional assumptions that constrain the supervised learning task, e.g., using task labels during inference.

These criteria are simple; however, most continual learning algorithms make strong assumptions that do not match real-world systems and are assessed on toy problems that are not appropriate surrogates for real-world problems where continual learning could greatly improve computational efficiency. For example, many works still focus on tasks such as permuted MNIST and split-CIFAR100 (Chaudhry et al., 2018b; 2019; Rahaf & Lucas, 2019; Pan et al., 2020; Titsias et al., 2019; Zenke et al., 2017; Rajasegaran et al., 2019), only work in extreme edge cases like incremental class learning (Castro et al., 2018; Chaudhry et al., 2018b; Hou et al., 2019; Rebuffi et al., 2017; Tao et al., 2020; Wu et al., 2019), assume the availability of task-labels during inference (Golkar et al., 2019; Fernando et al., 2017; Hung et al., 2019; Serra et al., 2018), or require large batches to learn (Yan et al., 2021; Douillard et al., 2022). For continual learning to have practical utility, we need efficiency and performance that rivals trained from scratch models, as well as robustness to data ordering.

There are two extreme frameworks for continual learning. At one extreme, is incremental batch learning where the agent receives a batch and has as much time as necessary to loop over that batch before proceeding to the next batch. Typically these systems are evaluated with large batches, and many experience dramatic performance decreases when smaller batches are used (Hayes et al., 2020). This setting is often studied in class incremental learning and domain incremental learning. At the other extreme is online learning, where the agent receives one input at a time that must be immediately learned. Humans and animals learn in a manner that is a compromise between these two extremes. They acquire new experiences in an online manner and these experiences are consolidated offline during sleep (McClelland & Goddard, 1996; Hayes et al., 2021). Sleep plays a role in memory consolidation in all animals studied, including invertebrates, birds, and mammals (Vorster & Born, 2015). While animals sleep to consolidate memories, they can use both consolidated (post-sleep) and recent (pre-sleep) experiences to make inferences while awake.

Although this paradigm is virtually ubiquitous among animals, it has rarely been studied as a paradigm in continual learning. This paradigm matches a real-world need: on-device continual learning and inference (Hayes & Kanan, 2022). For example, virtual/augmented reality (VR/AR) headsets could use continual learning to establish play boundaries and to identify locations in the physical world to augment with virtual overlays. Home robots, smart appliances, and smart phones need to learn about the environments and the preferences of their owners. In all of these examples, learning online is needed, but there are large periods of time where offline memory consolidation is possible, e.g., while the mobile device is being charged or its owner is asleep. In this paper, we formalize this paradigm for continual

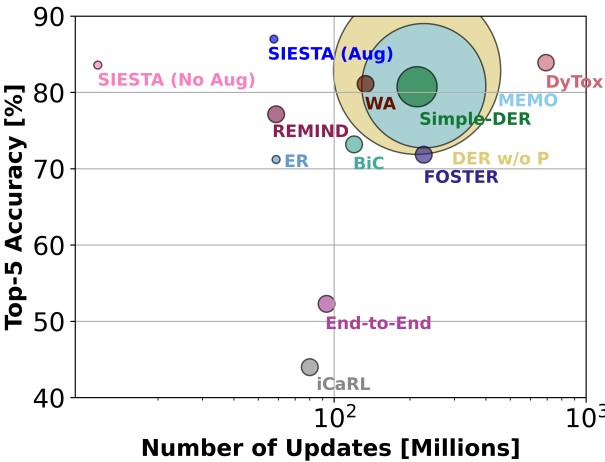

Figure 1: Our method, SIESTA, outperforms existing continual learning methods for class-incremental learning on ImageNet-1K while requiring fewer network updates and using fewer parameters, as denoted by circle size.

learning and we describe an algorithm with these capabilities, which we call SIESTA (**S**leep **I**ntegration for **E**pisodic **ST**re**A**ming).

For memory and computational efficiency, SIESTA adopts the quantized latent rehearsal scheme from RE-MIND (Hayes et al., 2020). REMIND continually trains the upper layers of a deep neural network (DNN) in a pseudo-online manner. It stores quantized mid-level representations of seen inputs in a buffer, which enables it to store a much larger number of samples with a given memory budget compared to veridical rehearsal methods that store raw images. To do pseudo-online training, REMIND uses rehearsal (Hetherington, 1989), a method for mitigating catastrophic forgetting by mixing new inputs with old inputs. For every new input, REMIND reconstructs a small number of past inputs, mixes the new input with them, and updates the DNN with this mini-batch; however, using rehearsal for every sample to be learned is not ideal. SIESTA addresses this by using rehearsal only during its offline sleep stage. For online learning, SIESTA instead uses lightweight online updates of the DNN's output layer.

**Our major contributions are summarized as:**

1. We formalize a framework for online updates with offline memory consolidation, and we describe the SIESTA algorithm that operates in this framework (see Figure 3). SIESTA is capable of rapid online learning and inference while awake, but has periods of sleep where it performs offline memory consolidation.

2. For incremental class learning on ImageNet-1K, SIESTA achieves state-of-the-art performance using far fewer parameters, memory, and computational resources than other methods. Without augmentations, training SIESTA requires only 1.9 hours on a single NVIDIA A5000 GPU. In contrast, recent methods require orders of magnitude more compute (see Figure 1). We confirm SIESTA's strong performance on three additional datasets.

3. SIESTA is the first continual learning algorithm to achieve identical performance to an offline model, when augmentations are not used. It suffers from zero *catastrophic forgetting in the augmentation-free setting* (see Table 1). SIESTA is capable of working with arbitrary orderings, and achieves similar performance in both class incremental and iid settings.

## 2 Online Updates with Offline Consolidation

We formalize the classification problem setting for supervised online continual learning with offline consolidation. The learner alternates between an online phase (wake) and an offline phase (sleep). For learning, during the $j$'th online phase, the agent receives a sequence of $n$ labeled observations, i.e., $t_{j1}, t_{j2}, \ldots, t_{jn}$, where each input observation

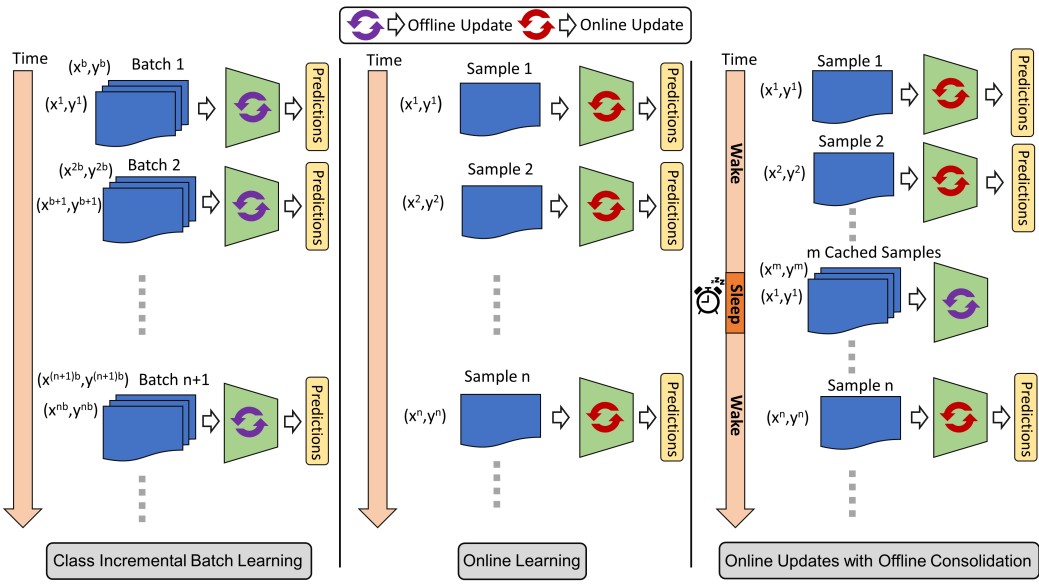

Figure 2: An overview of online updates with offline consolidation paradigm. While awake, agent performs online learning and while asleep, it performs computationally restricted offline learning. This wake/sleep cycles oscillate. Thus, our paradigm can be viewed as a combination of class incremental batch learning and online learning paradigms.

$x_t$ has label $y_t$. The sequence is not assumed to be stationary, and it can contain examples from classes from an arbitrary label ordering. The agent can cache these labeled pairs, or a subset of them, in memory with storage of size $b$ bits. The agent can be evaluated at any time during the online phase, where it must make inferences using both recent experiences from the $j$'th online phase, as well as past experiences from previous phases. During the subsequent offline consolidation phase, the agent is allowed at most $m$ updates of the network, e.g., gradient descent updates. We do not assume task labels are available during any phase. In general, iid (shuffled) orderings do not cause catastrophic forgetting; and at the other extreme, an ordering sorted by category causes severe catastrophic forgetting in conventional algorithms (Kemker et al., 2018).

With some exceptions (Kemker & Kanan, 2018; Pham et al., 2021; 2023; Arani et al., 2022), this paradigm has been little studied and offers several advantages. For example, it allows embedded mobile devices to quickly use new information from users and their environments and then consolidate that learning during a scheduled downtime. It also serves as a setting for studying learning efficiency in continual learning, where different sleep policies can be studied for minimizing the number of updates $m$ during a sleep setting. Lastly, it allows for testing functional hypotheses from neuroscience about sleep. While rehearsal-like mechanisms used in continual learning occur during slow wave sleep, the mechanisms that occur during rapid eye movement (REM) sleep have not yet been explored with DNNs nor has the interplay between slow wave sleep and REM sleep. REM sleep increases abstraction, facilitates pruning of synapses, and is when dreams occur (Smith & Smith, 2003; Djonlagic et al., 2009; Cai et al., 2009; Lewis et al., 2018; Durrant et al., 2015; Li et al., 2017). We juxtapose our training paradigm with alternatives in continual learning in Section 3.

Unlike incremental batch learning, in SIESTA compute is restricted during the sleep cycle. This enables us to explicitly model and control the amount of compute used during each rehearsal cycle. Traditional rehearsal is similar to our concept of sleep, where the model pauses after observing a certain number of samples to mix in new samples. An overview of this paradigm is juxtaposed with existing paradigms in Figure 2.

## 3 Related Work

We compare SIESTA's online updates with offline consolidation paradigm to alternative paradigms.

**Task Incremental Learning with Task Labels.** Incremental task batch learners (Kirkpatrick et al., 2017; Zenke et al., 2017; Aljundi et al., 2018; Chaudhry et al., 2018a;b; Serra et al., 2018; Dhar et al., 2019), learn from task batches, where each batch has a distinct task that is often a binary classification problem. These methods assume the task label is available during evaluation so that the correct "output" head can be selected, and when this assumption is violated these methods fail (Hayes & Kanan, 2020; Hayes et al., 2020). SIESTA does not require task labels for prediction, which are typically not available in real-world applications.

**Class Incremental Batch Learning.** In this paradigm, a dataset is split into multiple batches, where each batch consists of mutually exclusive categories, without any revisiting of categories. An agent is given a batch to learn for as long as it likes (see Figure 2) and typically can use some auxiliary memory for rehearsal. This paradigm has been studied with rehearsal-based methods (Hayes et al., 2021; Abraham & Robins, 2005; Belouadah & Popescu, 2019; Castro et al., 2018; Chaudhry et al., 2018b; French, 1997; Hayes et al., 2019; 2020; Hou et al., 2019; Rebuffi et al., 2017; Tao et al., 2020; Wu et al., 2019) that store previously observed data in a memory buffer or reconstruct them to rehearse alongside new data. It has also been studied in regularization based methods (Chaudhry et al., 2018b; Aljundi et al., 2018; Chaudhry et al., 2018a; Dhar et al., 2019; Kirkpatrick et al., 2017; Fernando et al., 2017; Coop et al., 2013; Li & Hoiem, 2017; Lopez-Paz & Ranzato, 2017; Ritter et al., 2018; Serra et al., 2018; Zenke et al., 2017) that constrain new weight updates to penalize large deviations from past weights, as well as dynamic methods (Douillard et al., 2022; Draelos et al., 2017; Yoon et al., 2018; Hou et al., 2018; Ostapenko et al., 2019; Rusu et al., 2016; Yan et al., 2021) that incrementally increase the capacity of a DNN over time. While task labels are not used during prediction, evaluation takes place between batches and many methods require large batches (e.g., thousands of examples) or they fail (Hayes et al., 2020). Some methods use a large number of parameters for keeping copies of the network in memory for distillation (Castro et al., 2018; Kang et al., 2022). We argue that while class incremental learning is a valuable assessment of a continual learner's ability to avoid catastrophic forgetting given an extremely adversarial data ordering, there is little real-world utility in algorithms that are designed solely to do class incremental learning. SIESTA differs from algorithms designed for this paradigm in that it can perform inference at any time and can operate for arbitrary data orderings, including when classes are revisited.

**Online Learning.** Unlike batch learning paradigms, in online learning, an agent observes data sequentially and learns them instance-by-instance in a single pass through the dataset (see Figure 2). To study catastrophic forgetting, examples are typically ordered by class, although alternative orders are sometimes studied (Hayes et al., 2020). Evaluation can occur at any point during training. This setting eliminates looping over data many times and evaluation between batches, thus making it more memory and compute time efficient, which is desirable for embedded devices. This paradigm has primarily been studied on smaller datasets (CIFAR-100) (Lopez-Paz & Ranzato, 2017; Chaudhry et al., 2018b; Rahaf & Lucas, 2019; Wang et al., 2021), although some methods have been shown to scale to ImageNet-1K (Hayes & Kanan, 2020; Hayes et al., 2019; Hayes & Kanan, 2020; Gallardo et al., 2021). For ImageNet-1K, these methods under-perform incremental batch learning methods (Hayes et al., 2020). SIESTA's paradigm is a compromise that captures most of the benefits of online continual learning while enabling increased accuracy.

**Paradigm Relationships.** The formal online updates with offline consolidation setting can be configured to mirror other continual learning settings (see Figure 2). For class incremental batch learning, where the learner receives large batches of training examples (e.g., $n > 100,000$ for ImageNet-1K (Rebuffi et al., 2017; Yan et al., 2021)), buffer $b = b_{recent} + b_{buffer} + b_{auxiliary}$ would be sufficiently large to hold all $n$ observations from the $j$'th online phase ($b_{recent}$) as well as past examples used for rehearsal ($b_{buffer}$), and any additional memory needed for other purposes ($b_{auxiliary}$) (e.g., distillation), and $m$ is typically very large (e.g., for the state-of-the-art method DyTox (Douillard et al., 2022), $m > 500n$). A pseudo-online algorithm like REMIND (Hayes et al., 2020) uses a configuration of $n = 1$ and $m = 51$. For both REMIND and SIESTA, $b$ only acts as a buffer for storing compressed past observations and their labels.

## 4 The SIESTA Algorithm

The SIESTA algorithm (Figure 3) alternates between awake and sleep phases. The awake phase involves online learning as well as sample compression, storage, and inference. The sleep phase involves memory consolidation via brief periods of offline learning. SIESTA is designed to handle data streams with arbitrary class orders, ranging from iid to class incremental paradigms.

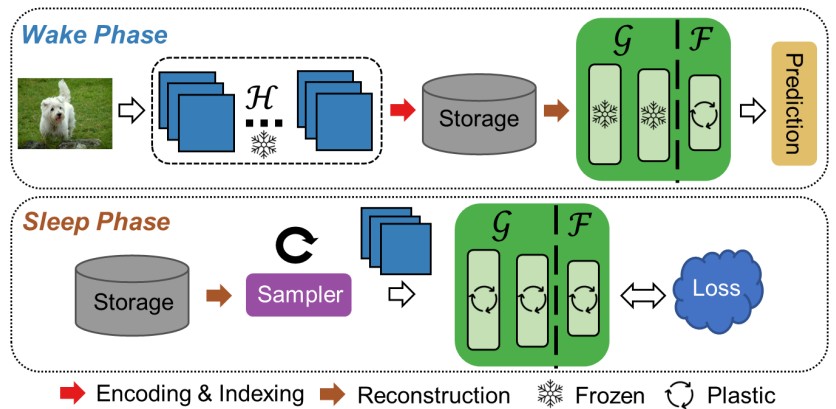

Figure 3: A high-level overview of SIESTA. During the *Wake Phase*, it transforms raw inputs into intermediate feature representations using network $\mathcal{H}$. The inputs are then compressed with tensor quantization and cached. Then, weights belonging to recently seen classes in network $\mathcal{F}$ are updated with a running class mean using the output vectors from $\mathcal{G}$. Finally, inference is performed on the current sample. During the *Sleep Phase*, a sampler uses a rehearsal policy to choose which examples should be reconstructed from the cached data for each mini-batch. Then, networks $\mathcal{G}$ and $\mathcal{F}$ are updated with backpropagation in a supervised manner. The wake/sleep cycles alternate.

SIESTA is a feed-forward DNN defined as $\mathcal{F}\left(\mathcal{G}\left(\mathcal{H}\left(\cdot\right)\right)\right)$, where $\mathcal{H}(\cdot)$ contains the bottom layers, $\mathcal{G}(\cdot)$ contains the top layers prior to the output layer, and $\mathcal{F}(\cdot)$ is the output layer. Specifically, SIESTA takes as input a 3rd-order tensor $\mathbf{X}$. $\mathcal{H}(\cdot)$ produces $\mathbf{Z} = \mathcal{H}\left(\mathbf{X}\right)$, where $\mathbf{Z} \in \mathbb{R}^{r \times s \times d}$, $r$ and $s$ are the tensor spatial dimensions, and $d$ are the tensor channel dimensions. This tensor is then transformed into a vector embedding, i.e., $\mathbf{z} = \mathcal{G}\left(\mathbf{Z}\right)$. The output layer $\mathcal{F}\left(\cdot\right)$ is then described with cosine softmax, where the score for the $k$'th class is given by:

$$p_k = \frac{\exp\left(a_k \tau^{-1}\right)}{\sum_j \exp\left(a_j \tau^{-1}\right)}, \tag{1}$$

where

$$a_k = \frac{\mathbf{f}_k^T \mathbf{z}}{\|\mathbf{f}_k\|_2 \|\mathbf{z}\|_2}, \tag{2}$$

$\mathbf{f}_k$ is the weight vector for the $k$'th class, and $\tau \in \mathbb{R}$ is a learned temperature used during optimization. It has been proven that cosine softmax encourages greater class separation than softmax (Kornblith et al., 2021). In our implementation, $\mathcal{H}$ and $\mathcal{G}$ are both convolutional networks; however, other architectural choices would be suitable.

Following Hayes et al. (2020), prior to continual learning, the DNN is initialized by pre-training on an initial set of $N$ training samples, e.g., images from the first 100 classes of ImageNet. Tensor features $\mathbf{Z}$ are extracted from each of the $N$ samples and used to fit a Product Quantization (PQ) model (Jegou et al., 2010) to all $rsN$ $d$-dimensional vectors in these tensors. This enables us to efficiently store and reconstruct compressed representations of $\mathbf{Z}$. This approach enables SIESTA to much more efficiently use memory for rehearsal than methods that store raw images. The network $\mathcal{H}\left(\cdot\right)$ is then kept fixed during continual learning. While this aspect of SIESTA is the same as REMIND, SIESTA differs significantly in its capabilities and how it is trained. REMIND uses rehearsal during online learning by sampling a minibatch that has 50 old examples and the currently observed example. Instead, SIESTA does not use rehearsal for its online updates and it only employs rehearsal during its sleep phase. We next describe how SIESTA's two learning phases operate.

## 4.1 Online Learning while Awake

During the awake phase (see Figure 3), only the output layer $\mathcal{F}$ of the DNN is updated. This enables SIESTA to avoid catastrophic forgetting and permits lightweight online updates. When SIESTA receives an input tensor $\mathbf{X}_t$ at time $t$, $\mathbf{Z}_t$ is then compressed and saved in a limited-sized storage buffer using PQ along with its class label. If the buffer is full, then a randomly selected sample is removed from the class with the most samples. Subsequently, the output layer

weights are updated with simple running updates for the appropriate class. The update for the output layer weight vector for class $k$ is given by

$$\mathbf{f}_k \leftarrow \frac{c_k \mathbf{f}_k + \mathbf{z}_t}{c_k + 1}, \tag{3}$$

where $c_k$ is an integer counter for class $k$. After updating the weight vector, $c_k$ is incremented, i.e., $c_k \leftarrow c_k + 1$. For inference, the class with the highest score $p_k$ in Equation 1 is selected as the predicted class.

## 4.2 Memory Consolidation During Sleep

During the sleep phase (see Figure 3), the output layer $\mathcal{F}$ and the top layers $\mathcal{G}$ are trained using rehearsal, while the bottom layers $\mathcal{H}$ are kept frozen. Rehearsal consists of selecting mini-batches of stored examples in the buffer for reconstruction and then using them to update the network with backpropagation. Following the paradigm in Section 2, the DNN is allowed at most $m$ gradient descent updates. Given a mini-batch of size $q$, each sleep cycle therefore consists of updating the DNN with $n$ mini-batches, where the total number of updates is $m = q \times n$. At the beginning of a sleep cycle, the samples chosen for reconstruction are governed by a policy. Our main results all use balanced uniform sampling, i.e., sampling an equal number from each class, which worked best on class balanced datasets and was competitive on long-tailed ones. Other policies are studied in Appendix D.

In a subset of our experiments, we use augmentation during learning. While augmentation is typically applied directly to images, here we apply it to the reconstructed $\mathbf{Z}$ tensors. We use two forms of augmentation: manifold mix-up (Verma et al., 2019) and cut-mix (Yun et al., 2019). Both strategies are used in the standard manner, except instead of producing a weighted combination of two images, we create a weighted combination of tensors.

In our main results, we have the network sleep every $120K$ samples, which in the incremental class learning setting corresponds to training on 100 categories. We study the impact of sleep frequency and sleep length in Section 6.3.

## 4.3 Network Architecture & Initialization

While continual learning is starting to use transformers (Douillard et al., 2022), recent work has primarily used ResNet18 (Rebuffi et al., 2017; Wu et al., 2019; Castro et al., 2018; Wu et al., 2019; Hayes et al., 2020; Yan et al., 2021). However, ResNet18 has been shown to perform worse than other similarly sized DNNs (Hayes & Kanan, 2022). Moreover, given one of the major applications of continual learning is on-device learning, using a DNN designed for embedded devices is ideal. Therefore, in our main results, we use **MobileNetV3-L** (Howard et al., 2019). MobileNetV3-L (5.48M) is lightweight with $2\times$ fewer parameters than ResNet18 (11.69M) and has lower latency. Since PQ encodes features across channels, MobileNetV3-L is more suitable for compressing its features with relatively less reconstruction error than ResNet18. We compare MobileNetV3-L and ResNet18 in Appendix E.

In Gallardo et al. (2021), pre-training (base initialization) with the self-supervised learning (SSL) algorithm SwAV (Caron et al., 2020) outperformed supervised pre-training for continual learning. We use their pre-training method in our main results, but we study other SSL methods in Appendix K. For fair comparisons, we use the same SwAV pre-trained DNN with DER (Yan et al., 2021), ER (Chaudhry et al., 2019), and REMIND (Hayes et al., 2020). Using MobileNetV3-L, we set network $\mathcal{H}$ to be the first 8 layers of the network, consisting of $2.19\%$ of the network parameters, which was found to be the best balance of accuracy and efficiency (see Appendix F). Using images of size $224 \times 224$ pixels, $\mathcal{H}$ produces a tensor $\mathbf{Z} \in \mathbb{R}^{14 \times 14 \times 80}$. For PQ, we use *Optimized Product Quantization (OPQ)* from FAISS (Johnson et al., 2019), which is used to compress and reconstruct the $14^2$ 80-dimensional vectors that make up each tensor. Following REMIND, we exclusively use reconstructed versions of the output of $\mathcal{H}$ during continual learning. The remaining 11 layers of MobileNetV3-L ($97.81\%$ of the DNN parameters) are trained during sleep.

# 5 Experimental Setup

**Comparison Models.** We compare SIESTA to a variety of baseline and state-of-the-art methods, including online learners REMIND (Hayes et al., 2020), ER (Chaudhry et al., 2019), SLDA (Hayes & Kanan, 2020), NCM (Mensink et al., 2013); incremental batch learners iCaRL (Rebuffi et al., 2017), BiC (Wu et al., 2019), End-to-End (Castro et al., 2018), WA (Zhao et al., 2020), PODNet (Douillard et al., 2020), Simple-DER (Li et al., 2021), DER (Yan et al., 2021), MEMO (Zhou et al., 2023b), FOSTER (Wang et al., 2022), DyTox (Douillard et al., 2022); and an offline learner. We

Table 1: Class incremental learning results on ImageNet-1K. For a fair comparison, we constrain methods to 12.5 million updates and do not use data augmentation. The ($\uparrow$) and ($\downarrow$) indicate high and low values to reflect optimum performance respectively. $\mathcal{P}$ is the number of parameters in Millions, $\mu$ is the average top-5 accuracy, $\alpha$ is the final top-5 accuracy, $\mathcal{M}$ is the total memory in GB, and $\mathcal{U}$ is the total number of updates in Millions.

| Method | $\mathcal{P}(\downarrow)$ | $\mu(\uparrow)$ | $\alpha(\uparrow)$ | $\mathcal{M}(\downarrow)$ | $\mathcal{U}(\downarrow)$ | GFLOPS ($\uparrow$) |
|---|---|---|---|---|---|---|
| Offline | 5.48 | — | 83.31 | 192.87 | 768.70 | — |
| DER | 54.80 | 81.87 | 70.15 | 20.99 | 12.43 | 7944.60 |
| ER | 5.48 | 76.32 | 63.92 | 19.59 | 11.53 | 1294.10 |
| REMIND | 5.48 | 81.77 | 74.31 | 2.02 | 11.53 | 10139.00 |
| **SIESTA** | **5.48** | **88.33** | **83.59** | **2.02** | **11.53** | **19326.00** |

compare with two variants of DER: DER without pruning (referred to as DER†) and DER with pruning (referred to as DER∗). These methods have been designed to be effective for incremental class learning on ImageNet-1K, and more details are in Appendix B. SIESTA is trained with cross-entropy loss and uses SGD as its optimizer with the OneCycle learning rate (LR) scheduler (Smith & Topin, 2019) during each sleep phase. It uses a higher initial LR in the last layer to help learn the new tasks and a lower LR in earlier layers to mitigate forgetting of previously learned information. For each sleep cycle, we use a batch size of 64, momentum 0.9, weight decay $1e$-5, and an initial LR of 0.2 for the last layer. LR is reduced in earlier layers by a layer-wise decay factor of 0.99. SIESTA uses the same universal setting i.e., same hyperparameters and same network configuration for all learning phases. Additional details of SIESTA and implementation details for other methods are provided in Appendix C.

**Datasets and Evaluation Criteria.** We use four datasets in our experiments. For our main results, we use **ImageNet ILSVRC-2012** (Russakovsky et al., 2015), which is the standard object recognition benchmark for testing a model's ability to scale. It has 1.28 million images uniformly distributed across 1000 categories. We study another large-scale scene recognition dataset, **Places365-Standard** which is a subset of Places-2 Dataset (Zhou et al., 2017). We also use a long-tailed version of Places-2 Dataset, **Places365-LT** to evaluate rehearsal policies (see Appendix D for details). Additionally, we also evaluate performance on the **CUB-200** (Wah et al., 2011) dataset in Appendix L. We configure the datasets in the continual iid setting and the class incremental learning setting, where data is ordered by class and images are shuffled within each class. For evaluation, we report the average accuracy $\mu$ over all steps $T$, where $\mu = \frac{1}{T}\sum_{t=1}^{T}\alpha_t$, with $\alpha_t$ referring to the accuracy at step $t$. We also report the final accuracy $\alpha$ for continual learning models. Note that $\alpha$ means best accuracy for offline models. We use top-5 accuracy for ImageNet-1K and top-1 accuracy for Places. Additionally, we measure the total number of parameters $\mathcal{P}$ (in millions) and the total memory $\mathcal{M}$ (in gigabytes) used by each model. We also measure the total number of updates $\mathcal{U}$, where an update consists of a backward pass on a single input.

# 6 Results

## 6.1 Continual Learners vs. The Offline Learner

We first compare SIESTA, ER, DER, and REMIND to an offline learner. For many real-world applications, a continual learner that performs significantly worse than an offline learner is unacceptable. Moreover, for these applications we cannot make assumptions about the class order distribution of the data stream; so for SIESTA, ER, and REMIND, we study both the continual iid and class incremental learning settings, which can be seen as extreme best case and extreme worst case scenarios respectively. DER, as designed, is only capable of class incremental learning. All compared methods use the same SwAV pre-trained MobileNetV3-L DNN on the first 100 classes of ImageNet-1K. The offline learner is a MobileNetV3-L trained from scratch.

We first show results where all models omit augmentations, including the offline learner. SIESTA used 1.28M updates per sleep cycle. All continual learners used a similar total number of updates. Results for class incremental learning on ImageNet-1K are given in Table 1 and learning curves in Figure 4a. DER, ER, and REMIND perform over 9% worse (absolute) than the offline learner for final accuracy. In contrast, SIESTA matches the offline learner's performance for final accuracy.

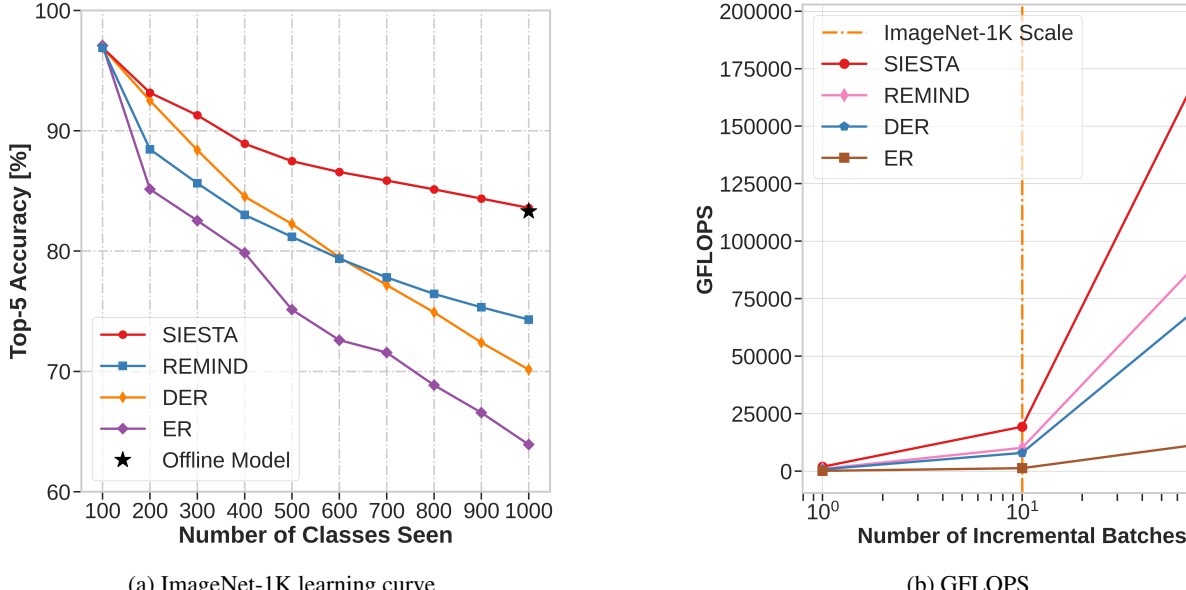

Figure 4: Comparison among SIESTA and baselines without augmentations. **(a)** We show learning curves on ImageNet-1K comparing continual learners with an offline learner. **(b)** We also compare continual learners based on GFLOPS. Each incremental batch corresponds to 100 classes from ImageNet-1K.

We analyzed SIESTA, ER, and REMIND under the continual iid setting and compare it against the class incremental setting. For the iid setting, SIESTA obtains 83.45% in final accuracy, whereas ER and REMIND attain 64.92% and 79.52% in final accuracy respectively. When switching from iid to the class incremental setting ER and RE-MIND decrease 1% (absolute) and 5.21% (absolute) in final accuracy respectively. In contrast, SIESTA maintains similar performance in both settings and shows robustness to data ordering. To further analyze SIESTA's performance across these distributions compared to an offline model, we used Chochran's Q test (Conover, 1999), a non-parametric paired statistical test that can be used for comparing three or more classifiers, and we found no significant difference among SIESTA's final accuracy for the iid and class incremental settings compared to the offline learner ($P = 0.08$). Therefore, SIESTA suffers from zero forgetting on ImageNet-1K in the augmentation-free setting by matching the performance of the offline MobileNetV3-L across orderings.

In augmentation experiments, SIESTA outperforms DER, ER, and REMIND by $15.18\%$, $15.78\%$, and $4.03\%$ (absolute) respectively in final accuracy (see Table 6). We also compare SIESTA with its awake-only variant and other online learning methods including REMIND, ER, SLDA, and NCM in Appendix H where SIESTA outperforms all compared methods and SIESTA (awake-only) shows competitive performance. SIESTA was found to be robust to data ordering in the augmentation setting as well. Using a McNemar's test on the final accuracy for SIESTA in the iid and class incremental settings revealed no significant difference ($P = 0.85$). However, with augmentations there was a significant difference between the offline learner and SIESTA based on a McNemar's test ($P < 0.001$). Additional details are in Appendix G. We hypothesize the offline model outperformed SIESTA due to the features learned in network $\mathcal{H}$ being not sufficiently universal (see Appendix K).

## 6.2 Computational & Memory Efficiency

A major goal of SIESTA is efficiency via continual learning, whereas the vast majority of continual learning systems are not more efficient than retraining an offline model (Harun et al., 2023a). To assess computational efficiency of models, we exclude the pre-training phase, since it only occurs once and our focus is enabling continual learning to be significantly more efficient than periodic retraining from scratch.

In terms of memory efficiency, REMIND and SIESTA use memory to store compressed tensors and the others use it to store 20,000 images. As shown in Table 1 and Table 2, SIESTA requires $10\times$ less memory than other methods

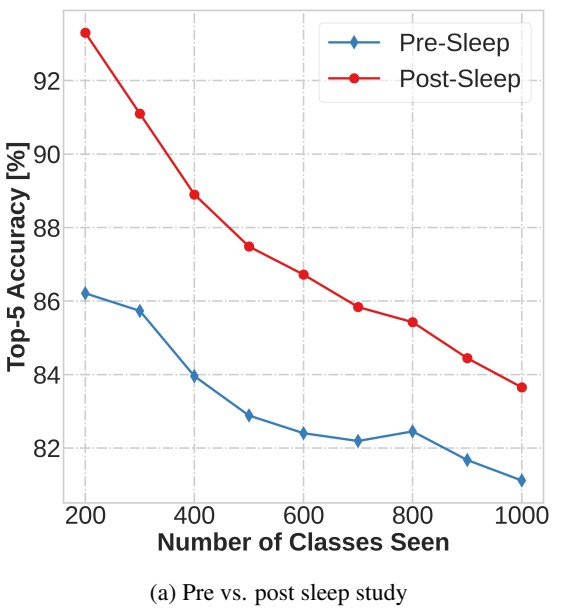
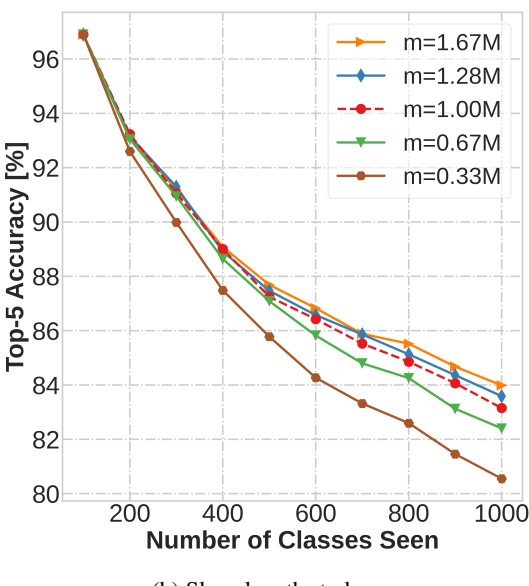

(a) Pre vs. post sleep study

(b) Sleep length study

Figure 5: Analysis of sleep in SIESTA (No Aug). **(a)** We study the overall impact of sleep using our default model configuration by showing the learning curves of pre- and post-sleep performances as a function of seen classes. **(b)** We also study the impact of sleep length $m$ on SIESTA's performance.

(19 − 22 gigabytes) except REMIND. Moreover, SIESTA requires $2\times - 20\times$ fewer parameters than most methods (11.68 − 116.89 million) (see Table 2). SIESTA is much more suitable for memory constrained real-world applications especially on-device learning than most other methods.

Based on the number of total DNN updates, SIESTA is much more computationally efficient than other methods. It requires $7\times - 60\times$ fewer DNN updates than others (see Table 2). We also empirically compared REMIND and SIESTA's continual learning training time when learning 900 ImageNet-1K classes using a single NVIDIA RTX A5000 GPU. Augmentations were not used with either model. REMIND required 8.1 hours. Using a batch size of 64 during sleep, which we used in the experiments in Section 6.1, SIESTA requires only 2.4 hours ($3.4\times$ faster than REMIND). **When the batch size is increased to 512, SIESTA required only 1.9 hours to continually learn 900 classes from ImageNet-1K with a negligible change in accuracy** (see Appendix J).

Using DeepSpeed[2] with the same GPU across models, we also conducted a computational analysis based on FLOPS (floating-point operations per second) which revealed that SIESTA has $1.9\times$, $2.4\times$, and $14.9\times$ higher GFLOPS than REMIND, DER, and ER respectively (see Table 1). Unlike other methods, SIESTA does not train the DNN during online learning and only performs a fixed number of backpropagation updates per sleep ($m = 1.28M$), thereby providing much higher GFLOPS than compared baselines. In real-world continual learning, we could potentially have an infinitely long (never ending) data stream, which would be larger than ImageNet-1K, raising the question: *how well do these models computationally scale to larger datasets?* In Figure 4b, we extrapolate from our FLOPS analysis to even larger datasets, which demonstrates that the gap in GFLOPS between SIESTA and other methods grows significantly, where SIESTA is predicted to be much more efficient than others in the large-scale dataset regime.

## 6.3 Sleep Analyses

We asked the question *"What is the impact of sleep on SIESTA's ability to learn and remember?"* These experiments are for incremental class learning, and data augmentation is not used. First, we analyze the pre-sleep and post-sleep performance of SIESTA on ImageNet-1K in Figure 5a using the same sleep settings as in Section 6.1. We see that the performance of SIESTA after sleep is consistently higher than before sleep for all increments, providing a $4.25\pm1.38\%$ average absolute increase in accuracy after each sleep cycle.

---

[2]https://github.com/microsoft/DeepSpeed

Table 2: Experimental results with state-of-the-art methods on ImageNet-1K. DER without pruning and DER with pruning are abbreviated as DER† and DER∗, respectively. The (↑) and (↓) indicate high and low values to reflect optimum performance respectively. $\mathcal{P}$ is the number of parameters in millions, $\mu$ is the average top-5 accuracy, $\alpha$ is the final top-5 accuracy, $\mathcal{M}$ is the total memory in GB, and $\mathcal{U}$ is the total number of updates in millions. We include SIESTA with (Aug) and without (No Aug) augmentations. DER∗ does not report $\mathcal{P}$, so we omit it.

| Method | $\mathcal{P}(\downarrow)$ | $\mu(\uparrow)$ | $\alpha(\uparrow)$ | $\mathcal{M}(\downarrow)$ | $\mathcal{U}(\downarrow)$ |
|---|---|---|---|---|---|
| End-to-End (Castro et al., 2018) | 11.68 | 72.09 | 52.29 | 22.32 | 93.26 |
| Simple-DER (Li et al., 2021) | 28.00 | 85.62 | 80.76 | 22.39 | 213.17 |
| iCaRL (Rebuffi et al., 2017) | 11.68 | 63.70 | 44.00 | 22.32 | 79.94 |
| BiC (Wu et al., 2019) | 11.68 | 84.00 | 73.20 | 22.32 | 119.91 |
| WA (Zhao et al., 2020) | 11.68 | 86.60 | 81.10 | 22.32 | 133.23 |
| PODNet (Douillard et al., 2020) | 11.68 | 81.93 | 71.39 | 22.32 | 226.49 |
| DER† (Yan et al., 2021) | 116.89 | 88.17 | 82.86 | 22.74 | 213.17 |
| DER∗ (Yan et al., 2021) | — | 87.08 | 81.89 | 22.32 | 213.17 |
| MEMO (Zhou et al., 2023b) | 86.72 | 87.15 | 80.89 | 22.65 | 226.49 |
| FOSTER (Wang et al., 2022) | 11.68 | 83.23 | 71.85 | 22.32 | 226.49 |
| DyTox (Douillard et al., 2022) | 11.36 | 88.78 | 83.91 | 22.32 | 692.80 |
| ER (Chaudhry et al., 2019) | 5.48 | 82.19 | 71.22 | 19.59 | 58.78 |
| REMIND (Gallardo et al., 2021) | 11.68 | 83.61 | 77.14 | 2.05 | 58.78 |
| **SIESTA** (No Aug) | 5.48 | 88.33 | 83.59 | 2.02 | **11.53** |
| **SIESTA** (Aug) | **5.48** | **90.67** | **87.00** | **2.02** | 57.60 |

Next, we study the impact of sleep frequency. To do this, we trained models with a sleep frequency of 50, 100 (default), or 150 classes seen. For this analysis, we used a fixed sleep length, i.e., a fixed number of updates per sleep ($m = 1.28M$). The absolute average increase in post-sleep accuracy was $2.29 \pm 0.86\%$ for 50 classes, $4.25 \pm 1.38\%$ for 100 classes, and $6.18 \pm 2.15\%$ for 150 classes. Despite the 50 class increment being trained with more updates, the 100 class increment achieved better post-sleep performance. We hypothesize that this happens because frequent sleep leads to greater perturbation in the DNN's weights, resulting in gradual forgetting of old memories.

In Figure 5b, we study the impact of sleep length on SIESTA's performance by varying the number of updates ($m$) during each sleep period, where SIESTA slept every 100 classes. We observe that as sleep length increases, SIESTA's performance also increases; however, as the sleep length increases, SIESTA requires more updates and there are diminishing returns in terms of increases in accuracy, so we must strike a balance between accuracy and efficiency.

## 6.4 State-of-the-Art Comparisons

To put our work in context with respect to existing methods, we compare SIESTA against recent class incremental learning methods that have previously been shown to perform well on ImageNet-1K. All methods, except for SIESTA (No Aug), use augmentations and have a variety of different DNN architectures. SIESTA (Aug), which uses augmentations, used 6.4 million updates per sleep cycle ($m = 6.4M$). With the exception of REMIND, ER, and SIESTA, we use published performance numbers for all methods. REMIND and SIESTA both use around 2GB of memory for rehearsal and SwAV for pre-training. ER uses same SwAV pre-trained DNN as SIESTA. Additional details for the comparison algorithms are provided in Appendix B.

Overall results are in Table 2 and Figure 1. SIESTA performs best in average accuracy $\mu$ and final accuracy $\alpha$, while having fewer DNN parameters $\mathcal{P}$, lower auxiliary memory usage $\mathcal{M}$, and fewer total updates $\mathcal{U}$. While REMIND uses the same amount of auxiliary memory and a similar number of DNN updates, SIESTA (Aug) outperforms REMIND by $9.86\%$ (absolute) in final accuracy. SIESTA (Aug) exceeds DyTox, the method with the second highest final accuracy, by $3.09\%$ (absolute), while using $12\times$ fewer network updates and $11\times$ less memory. SIESTA (No Aug) has comparable performance to state of the art batch learners like DER and DyTox, while requiring $18\times$ fewer updates. Moreover, SIESTA (Aug) can provide even further performance gains ($3.41\%$ absolute improvement in final accuracy) at the cost of $5\times$ more updates.

## 6.5 Additional Experiments & Ablation Studies

We conduct ablation experiments over a number of facets of SIESTA in order to gain insight into their relative importance. Details are in the Appendix.

**Rehearsal Policies.** In Appendix D, we compared eight different rehearsal policies. To do this, we used both ImageNet-1K and a challenging long-tailed dataset, Places365-LT. On ImageNet, most policies performed similarly; however, large differences were seen among methods for Places365-LT, where class balanced uniform sampling outperforms alternatives. SIESTA with or without balanced uniform rehearsal outperforms REMIND by $3.15\%$ (uniform) and $7.88\%$ (balanced uniform) in final accuracy on Places365-LT. SIESTA achieves $4.73\%$ higher final accuracy on Places365-LT when using balanced uniform rehearsal compared to uniform rehearsal.

**Architectures.** We compare MobileNetV3-L and ResNet18 in Appendix E where MobileNetV3-L outperforms ResNet18 by $3.84\%$ (augmentation) and $5.03\%$ (no augmentation) in accuracy on ImageNet-1K.

**Buffer Size.** We study the impact of buffer size on SIESTA's performance in Appendix I where reducing buffer size from 2GB to 0.75GB results in only $3.02\%$ absolute drop in final accuracy. Additionally, when both SIESTA and DER store same number of samples in buffer (130000 samples), SIESTA achieves $70.14\%$ final accuracy on ImageNet-1K compared to DER's $70.15\%$. However, DER requires $10\times$ more parameters and $95.4\times$ more memory than SIESTA. When another variant of SIESTA, SIESTA-ER stores 130000 raw images like DER and performs veridical rehearsal, it outperforms DER with $71.86\%$ final accuracy on ImageNet-1K. Furthermore, unlike DER, SIESTA is able to rival the offline model with a low memory footprint (see Table 1 and Table 6).

**Does SIESTA Require Latent Rehearsal?** For memory-efficiency, SIESTA uses latent rehearsal. To examine if SIESTA's innovations apply to the veridical rehearsal setting, where raw images are used, we created a veridical rehearsal variant of SIESTA. Compared to ER (Chaudhry et al., 2019), a widely used veridical rehearsal method, this variant of SIESTA (SIESTA-ER) outperforms ER by $7.94\%$ (absolute) in final accuracy on ImageNet-1K (see Figure 7), while being $3\times$ faster to train than ER on same hardware.

**Additional Datasets.** Besides ImageNet-1K, we evaluated SIESTA on three additional datasets e.g., Places365-LT, Places365-Standard, and CUB-200 in Appendix D and Appendix L. SIESTA achieves highest accuracy in all experiments compared to other methods. In particular, SIESTA outperforms ER and REMIND by $17.17\%$ and $7.55\%$ (absolute) in final accuracy on the Places365-Standard dataset (Table 10). SIESTA also outperforms ER and REMIND by $8.9\%$ and $12.76\%$ (absolute) respectively in final accuracy on the CUB-200 dataset (Table 11). SIESTA learns the large-scale Places365-Standard dataset (1.8M training samples) $4.4\times$ faster than REMIND using the same hardware.

## 7 Discussion

This paper considers the problem of supervised continual learning, where the learner incrementally learns from a sequence about which it cannot make distributional assumptions. We argue that for real-world applications, continual learning needs to rival an offline learner and be more computationally efficient than periodically re-training from scratch. We also argue that systems need to be designed to handle arbitrary class orderings, where two extremes are iid and class-incremental learning, whereas many continual learning systems are bespoke to the incremental class learning scenario. We demonstrated that SIESTA largely meets these goals, achieving identical performance to the offline learner when augmentations are not used, and outperforming existing continual learning methods in accuracy using less compute when augmentations are used.

Computational efficiency in continual learning has long been a selling point of the research, but it has received little attention. Training large DNNs from scratch requires a huge amount of energy that can result in a large amount of greenhouse gas emissions (Luccioni et al., 2022; Patterson et al., 2021; Wu et al., 2022). From a financial perspective, training large models often requires a large amount of electricity, expensive hardware, and cloud computing resources. Many are calling for more computationally efficient algorithms to be developed (Strubell et al., 2020; Van Wynsberghe, 2021; Harun & Kanan, 2023), and we believe that continual learning can help address this problem while also enabling greater functionality provided by continuously updating DNNs with new information.

In this work, we only evaluated CNN architectures with SIESTA; however, the general framework is amenable to other architectures as long as the representations can be quantized. For example, SIESTA could be extended for

models like Swin Transformers (Liu et al., 2021) or even Graph Neural Networks (Zhou et al., 2020), where we could quantize graph representations. Exploring the use of non-CNN architectures is critical for using SIESTA in non-vision modalities, e.g., audio, which would be an exciting area of future work. Another interesting area of research could be incorporating additional data modalities over time to improve existing task performance and enhancing the learning of new tasks. Another direction is to use SIESTA for tasks such as continual learning in object detection, which was previously done using a REMIND-based system (Acharya et al., 2020). It would also be interesting to study SIESTA's behavior on multi-modal tasks that incorporate vision and language (Kafle et al., 2019), although it is not clear how necessary rehearsal is for text-based continual learning (Ke et al., 2023).

The SIESTA model depends on the features learned in initial layers of the network, $\mathcal{H}$, being universal features for the domain, since they are not trained after the base initialization phase. Following previous continual learning works (Hayes et al., 2020; Belouadah & Popescu, 2019; Gallardo et al., 2021), we did base initialization on the first 100 classes of ImageNet-1K. While our model rivaled the offline learner when augmentations were not used, there was a small gap when augmentations were used between SIESTA and the offline learner. We hypothesize that this gap would be closed by improving the features in $\mathcal{H}$. Two potential ways to achieve this goal would be using a superior self-supervised learning algorithm than SwAV or by training on additional data. For continual learning for real applications, it would be prudent to initialize the DNN from a very large unlabeled dataset with self-supervised learning, which would likely work significantly better.

There are multiple approaches that could be explored to improve SIESTA's sample efficiency to reduce the length of sleep cycles. In Harun & Kanan (2023), it was shown that adjusting the training process to control plasticity could lead to significant improvements in transfer across time with increased sample efficiency. In this work we randomly sampled the rehearsal buffer during the sleep stage, but it has recently been shown that smart rehearsal policies can reduce the number of samples needed for rehearsal in SIESTA and other methods (Harun et al., 2023b).

In real-world settings, a data stream may have label noise. This problem has been little studied in existing CL research, it would be an interesting future direction to study in the SIESTA framework. One potential way this could be done is by using selective sampling of rehearsal samples to mitigate the effects of label noise, which was studied in Kim et al. (2021) and Karim et al. (2022) for small-scale datasets.

## 8 Conclusion

We proposed SIESTA, a scalable, faster and lightweight continual learning framework equipped with offline memory consolidation. We reduced computational overhead by making online learning free of rehearsal or expensive parametric updates during the wake phase. This closely aligns with real-time applications such as edge-devices, mobile phones, smart home appliances, robots, and virtual assistants. To effectively learn from a non-stationary data stream, the short-term wake memories were transferred into the DNN for long-term storage during offline sleep periods. Consequently, our model overcame forgetting of past knowledge. We showed that sleep improved online performance while outperforming state-of-the-art methods. SIESTA achieves similar accuracy to DyTox (Douillard et al., 2022) using an order of magnitude fewer updates when augmentations are not used, and surpasses DyTox in terms of accuracy when using augmentations.

### Acknowledgments

We thank Robik Shrestha for providing comments on the manuscript. This work was supported in part by NSF awards #1909696, #2326491, and #2125362. The views and conclusions contained herein are those of the authors and should not be interpreted as representing the official policies or endorsements of any sponsor.

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

# Appendix

## A   Implementation Details and Additional Results

We organize additional supporting experimental findings as follows: Appendix B contains brief descriptions of the compared methods. The implementation details about SIESTA, REMIND, DER, and ER are included in Appendix C. An analysis on rehearsal policies is located in Appendix D. We compare offline MobileNetV3-L with offline ResNet18 in Appendix E, and we analyze MobileNet quantization layers in Appendix F. A comparison with the offline model when augmentations are used can be found in Appendix G. Appendix H describes online learning results when sleep is omitted. For SIESTA, we study the impact of buffer size in Appendix I and batch size in Appendix J. We include SSL pre-training analysis in Appendix K. Finally, we include experimental results on Places365-Standard and CUB-200 in Appendix L.

## B   Comparison Algorithms

This paper compared SIESTA's performance against state-of-the-art continual learning algorithms, including:

- **REMIND** (Hayes et al., 2020): Instead of veridical rehearsal (i.e., raw pixels), REMIND performs latent rehearsal using mid-level CNN features that are compressed by PQ. This significantly reduces memory and enables REMIND to store more examples to mitigate catastrophic forgetting. It keeps earlier CNN layers fixed for feature extraction and trains later plastic layers for classification. In Gallardo et al. (2021), it was shown that replacing supervised pretraining by self-supervised pretraining improves REMIND's performance.

- **SLDA** (Hayes & Kanan, 2020): SLDA learns only the final classification layer while keeping earlier pretrained CNN layers fixed. It stores separate mean vectors for each class and a shared covariance matrix; only these class statistics are updated during online training and used for predictions.

- **iCaRL** (Rebuffi et al., 2017): iCaRL performs incremental batch learning using veridical rehearsal. It utilizes distillation loss and a nearest class mean classifier to prevent catastrophic forgetting.

- **BiC** (Wu et al., 2019): BiC builds on iCaRL and also uses distillation loss and rehearsal. It introduces a bias correction mechanism to prevent bias from class imbalance. For that, a linear model is learned on validation data to recalibrate the model's probabilities.

- **WA** (Zhao et al., 2020): WA applies a knowledge distillation loss and a bias correction step where it aligns the norms of new class weights to those of old class weights.

- **End-to-End** (Castro et al., 2018): End-to-End is a variant of iCaRL. Instead of the nearest class mean classifier, it utilizes the CNN's output layer.

- **PODNet** (Douillard et al., 2020): PODNet utilizes a spatial-based distillation loss to preserve representations throughout the model.

- **Experience Replay (ER)** (Chaudhry et al., 2019): ER maintains a constrained memory buffer consisting of a subset of old images (raw pixels) which is combined with newly arrived images to update the whole network.

- **DER** (Yan et al., 2021): By reusing previous feature extractors, DER freezes previously learned representations and augments them with incoming features obtained from a newly trained feature extractor. It employs a channel-level mask-based pruning mechanism to dynamically expand representations. It uses an auxiliary classifier along with a regular classifier to discriminate between old and new concepts. In this paper, we compare with two variants of DER: DER without pruning (referred to as DER†) and DER with pruning (referred to as DER∗). We also compare with a previous version referred to as Simple-DER (Li et al., 2021) which also combines multiple feature extractors and applies pruning. However, Simple-DER still has $2.4\times$ more parameters than offline ResNet18.

- **MEMO** (Zhou et al., 2023b): Instead of expanding entire network like DER, MEMO only expands special-ized blocks corresponding to the deep layers in the network. Thus MEMO reduces memory footprint for dynamic model expansion.

- **FOSTER** (Wang et al., 2022): Since saving an entire network per task is memory intensive, FOSTER adds an additional model compression technique using knowledge distillation.

- **DyTox** (Douillard et al., 2022): DyTox is a Transformer-based encoder/decoder framework where the encoder and decoder are shared among tasks. Unlike DER which keeps a copy of the whole network per task, DyTox proposes a dynamic task token expansion method to adapt to new tasks. Thus it alleviates the issues of a) expanding the network parameters to scale and b) requiring task identifiers in dynamic expansion methods.

- **Nearest Class Mean (NCM)** (Mensink et al., 2013): NCM computes a running mean for each class. During test time, a test sample is given the label of the nearest class mean. It becomes a variant of SLDA if the covariance matrix is fixed to the identity matrix.

- **Offline**: The offline model has access to the entire dataset and is trained using conventional batch training with multiple epochs. It serves as an approximate upper bound for a continual learning model.

Veridical rehearsal methods apply various image augmentations and these differ across these systems. For example, BiC and DER use random crops and horizontal flips, while DyTox uses Rand-Augment. REMIND uses feature augmentation by doing random crops and mixup directly on the stored mid-level features. To compare continual learning methods, rather than augmentation strategies, our experiments in Section 6.1 omitted augmentations during the continual learning phase for SIESTA, REMIND, ER, and DER. This allowed us to compare these algorithms in a fair way, where we also used the same DNN architecture.

In Table 2, we report results for REMIND and ER based on our implementations. For iCaRL, BiC, and WA, we report results from Yan et al. (2021). For PODNet, MEMO, and FOSTER, we report results from Zhou et al. (2023a). For other methods, we report results from their respective papers.

## C  Additional Implementation Details

### C.1  MobileNetV3-L

We use a slightly modified version of MobileNetV3-L with SIESTA, ER, DER, and REMIND. We use the GELU activation instead of ReLU and Hard Swish in all MobileNetV3-L layers, except for the squeeze and excitation block. We replace batch normalization with group normalization and weight standardization in the $\mathcal{F}$ and $\mathcal{G}$ layers. We used GELU activation because it improves gradient flow. And, we used group normalization with weight standardization because they are effective for small batch-sizes while performing competitively to batch normalization for large batch-sizes (Qiao et al., 2019). We incorporate these architectural modifications into MobileNetV3-L to get rid of adverse effects of batch normalization and make the model more general purpose.

The offline MobileNetV3-L was trained for 600 epochs using the AdamW optimizer with an initial LR of $0.004$ and a weight decay of $0.05$. We used the Cosine Annealing LR scheduler with a linear warm up of $5$ epochs.

### C.2  Base Initialization

SIESTA, REMIND, ER, and DER are initialized using all images in the first batch of 100 classes from ImageNet-1K, which is referred to as "base initialization." We use the same data ordering as REMIND (Hayes et al., 2020). Base initialization consists of three phases:

1. Following Gallardo et al. (2021), we first perform self-supervised pre-training of the network using SwAV. We use the small batch procedure from the original SwAV paper (Caron et al., 2020). Following others who have used self-supervised learning with mobile DNNs, we modified the scale of global and local crops in multi-crop augmentation. Following Tan et al. (2022), we use $2 \times 224 + 6 \times 128$ crops with a minimum scale crop range of $[0.3 - 0.05]$ and maximum scale crop range of $[1.0 - 0.3]$. We use 1000 prototypes, a Sinkhorn

regularization parameter $\epsilon$ of $0.03$, queue length of $384$ features, and $2000$ epochs. We set the base LR to $0.6$, final LR to $0.0006$, and batch size to $64$. The remaining settings are identical to the original SwAV (Caron et al., 2020) system.

2. For REMIND and SIESTA only, we then fit the parameters of the product quantization algorithm OPQ to the embeddings produced by $\mathcal{H}$. OPQ is configured to use $8$ codebooks of size $256$. In our main results, $\mathcal{H}$ consists of the first $8$ layers of the DNN.

3. Supervised fine-tuning is then done on the DNN, where for ER and DER the entire network is trained, but for REMIND and SIESTA only $\mathcal{F}$ and $\mathcal{G}$ are trained using the reconstructed tensors from $\mathcal{H}$. For ER and DER, we use the SGD optimizer with an initial LR of $0.1$ which is decayed by a $0.1$ multiplier every $15$ epochs. We use $50$ epochs and the standard PyTorch augmentations (random resized crop and random horizontal flip) for training ER and DER. For SIESTA and REMIND, we train for $50$ epochs. SGD is used for optimization with an initial LR of $0.2$ for the last layer. For remaining earlier layers, LR is reduced by a layer-wise decay factor of $0.99$. The LR is decayed by a $0.1$ multiplier every $15$ epochs. REMIND uses augmentations found optimum in the original REMIND paper (Hayes et al., 2020). SIESTA uses Cutmix and Mixup. These augmentations are applied using participation probability $p_{cutmix} = 0.6$ and $p_{mixup} = 0.4$. We set $\beta = 1.0$ for Cutmix and $\alpha = 0.1$ for Mixup.

After base-initialization, ER/DER achieves $96.90\%$ top-5 accuracy and REMIND/SIESTA achieve $96.84\%$ top-5 accuracy on the validation set for the first 100 classes.

### C.3   SIESTA

SIESTA's settings for training during continual learning are described in Section 5.

### C.4   REMIND

In our augmentation-free experiments, we configured REMIND to use the same number of updates as SIESTA ($11.53$M), which was done by setting REMIND to use a rehearsal mini-batch size of $9$.

In our augmentation experiments, we used REMIND's default configuration, which uses a rehearsal mini-batch size of $50$. For these experiments, REMIND uses the same augmentation scheme as in the original REMIND paper.

### C.5   DER

For our experiments without augmentations, we configured DER to use a similar number of updates as SIESTA and REMIND. To do this, we set the number of epochs used during each $100$ class increment to $10$. We adjusted the LR scheduler accordingly. In experiments with augmentations, DER used $47$ epochs in every $100$ class increment and random resized crop and random horizontal flip augmentations. Other implementation details e.g., optimizer and fine-tuning of the unified classifier follow the original DER paper (Yan et al., 2021). In all experiments, DER stored 10 MobileNetV3-L models for 10 learning steps, the current batch of at least 120000 images, and a rehearsal buffer of 10000 images ($224 \times 224$ uint8); all of which required $20.99$ GB in memory.

### C.6   ER

Unlike original ER (Chaudhry et al., 2019) that performs online learning in batch-by-batch manner i.e., learner receives a batch of new data at each time step, our ER updates whole network in sample-by-sample manner. For memory constraints, ER stores same number of raw images in the buffer as DER (130000). For compute constraints, ER rehearses the same number of instances as REMIND. ER uses SGD optimizer with a fixed LR of $0.1$. We omit LR scheduler since it degrades accuracy. Other settings such as layer-wise LR decay, momentum, and weigh decay follow SIESTA. In experiments with augmentations, ER uses random resized crop and random horizontal flip augmentations.

Table 3: Places365-LT experimental results without augmentation. Symbols (↑) and (↓) indicate high and low values to reflect optimum results. $\mu$ (average top-1 accuracy), $\alpha$ (final top-1 accuracy), $\mathcal{U}$ (total number of updates in Million). We show the mean $\pm$ standard deviation across 5 different class incremental orderings. SIESTA† is a variant with uniform sampling, while SIESTA is our main method with balanced uniform sampling.

| Methods | $\mu(\uparrow)$ | $\alpha(\uparrow)$ | $\mathcal{U}(\downarrow)$ |
|---------|-----------------|--------------------|---------------------------|
| Offline | — | 22.49 | 2.50 |
| REMIND | $25.98 \pm 1.10$ | $17.00 \pm 0.54$ | 0.28 |
| SIESTA† | $29.08 \pm 1.30$ | $20.15 \pm 0.41$ | **0.23** |
| **SIESTA** | $\mathbf{33.35 \pm 1.22}$ | $\mathbf{24.88 \pm 0.46}$ | 0.23 |

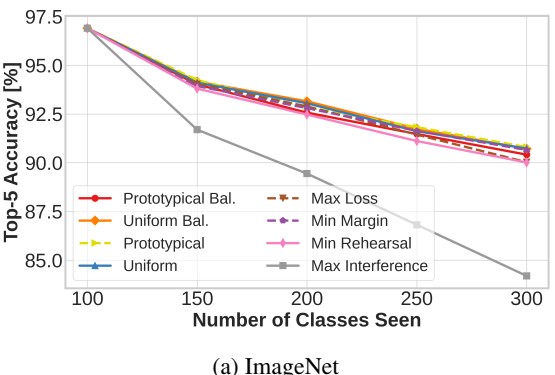

(a) ImageNet

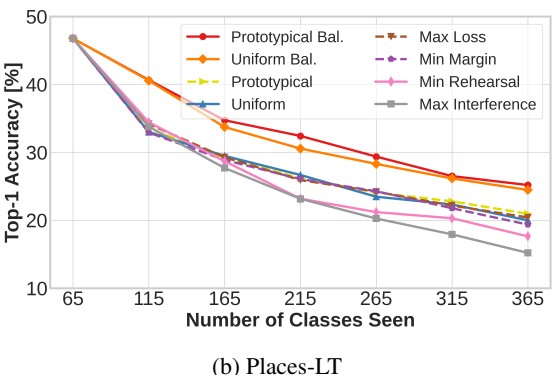

(b) Places-LT

Figure 6: Rehearsal policy analysis on: (a) ImageNet 300 classes (b) Places-LT 365 classes.

# D   Rehearsal Policies

Since real-world data distribution is often long-tailed (LT) and imbalanced, we investigate the robustness of SIESTA and REMIND to LT data distribution using Places365-LT dataset (Liu et al., 2019). It is a long tailed variant of the Places-2 dataset (Zhou et al., 2017). It has a total of 365 categories with 62500 training images raging from 5 to 4980 images per class. We use the Places365-LT validation set from Liu et al. (2019) as our test set consisting of a total of 7300 images with a balanced distribution of 20 images per class.

Since Places365-LT is a small dataset, we first initialized all MobileNetV3-L layers with SwAV weights pre-trained on ImageNet base initialization subset i.e., 100 classes. Next we fine-tuned the $\mathcal{F}$ and $\mathcal{G}$ layers using Places365-LT base initialization subset i.e., 65 randomly chosen classes. The fine-tuning phase uses the same configuration as our ImageNet experiments. After that, the remaining 300 classes are learned incrementally in a total of 6 steps where each step consists of 50 classes. SIESTA uses 38400 updates per sleep cycle and REMIND uses 5 rehearsal samples. The total number of updates for end-to-end training is bounded by 0.28M. The Offline model is initialized with SwAV weights and trained for 40 epochs using the same configuration as the ImageNet fine-tuning phase. Other implementation details follow ImageNet experiments with augmentation-free settings.

Table 3 compares SIESTA with REMIND in terms of mean $\pm$ standard deviation across 5 different random class orderings on Places365-LT. SIESTA† uses uniform sampling, while SIESTA uses uniform balanced sampling (balanced per class). We see that SIESTA† outperforms REMIND in terms of final and average accuracy and shows greater robustness than REMIND to Places365-LT. Additionally, we see that the final accuracy has a low standard deviation ($< 1\%$), demonstrating that REMIND and SIESTA† are robust against different data orderings. Also, we see that SIESTA with balanced uniform sampling (SIESTA) provides the highest accuracy, demonstrating its robustness to class imbalance.

After validating that balanced uniform sampling was superior to uniform sampling on Places365-LT, we studied additional rehearsal sampling strategies on both balanced (ImageNet-1K) and long-tailed (Places365-LT) distributions. For these experiments only 300 classes from ImageNet were used, where 200 were learned continually. We evaluated the following sampling policies:

Table 4: Offline comparison training only $\mathcal{F}$ and $\mathcal{G}$ on quantized features from $\mathcal{H}$ for MobileNetV3-Large and ResNet18 on ImageNet-1K. The ($\uparrow$) and ($\downarrow$) indicate high and low values to reflect optimum results respectively. $\mathcal{P}$ denotes the number of trainable parameters in $\mathcal{F}$ and $\mathcal{G}$ in millions. $\alpha$ and $\mathcal{E}_{OPQ}$ indicate best top-5 accuracy (%) and OPQ quantization error respectively.

| Architecture | Augmentation | $\mathcal{P}(\downarrow)$ | $\alpha(\uparrow)$ | $\mathcal{E}_{OPQ}(\downarrow)$ |
|---|---|---|---|---|
| ResNet18 | ✔ | 10.08 | 82.63 | 0.40 |
| MobileNetV3-L | ✔ | **5.36** | **86.47** | **0.18** |
| ResNet18 | ✗ | 10.08 | 79.54 | 0.40 |
| MobileNetV3-L | ✗ | **5.36** | **84.57** | **0.18** |

- *Balanced Uniform*: Sampling each class uniformly, so that each has the same number of samples. This is used in our main results.
- *Uniform*: Sampling uniformly regardless of category.
- *Min Rehearsal*: Examples are prioritized based on how many times they are rehearsed to update DNN parameters so that the DNN does not forget about the examples with least rehearsal counts.
- *Max Interference*: Samples having different class boundaries may interfere with one another due to the similarity or proximity in embedded space. These interfered samples may provide optimum supervision to improve performance. Interference is computed based on cosine similarity among penultimate features.
- *Max Loss*: Samples that the network is most uncertain about may better optimize the training objective. One way to implement this is by prioritizing samples that have the highest cross-entropy loss.
- *Min Margin*: The minimum margin policy prioritizes samples based on the separation between the highest and second highest probabilities given by the DNN. Lower separation corresponds to higher uncertainty and higher sampling probability.
- *Prototypical*: Samples closest to the respective class means are the most prototypical (i.e., class representative) whereas samples furthest from class means are the least prototypical. Giving higher priority to the most representative examples and lower priority to the least representative examples may improve learning that class.
- *Balanced Prototypical*: Selects equal number of samples per class using the prototypical criterion.

Since the standard deviation across orderings is low (see Table 3), we compare different policies based on a single ordering.

Results with different sampling policies are summarized in Figure 6. Only the balanced uniform sampling policy was effective for both datasets. On ImageNet (Figure 6a), we found that most of the rehearsal methods perform similarly except max interference, which performs poorly. Max interference rehearses hard examples which is detrimental for small models (Sorscher et al., 2022). In contrast, on Places365-LT (Figure 6b) we observed that balanced prototypical and balanced uniform performed best. We selected balanced uniform for our main results due to its simplicity and because it was effective across dataset label distributions.

## E    Architecture Comparisons

The goal of this study is to find an optimum network architecture for efficient continual learning. Here we compare MobileNetV3-Large with ResNet18 to examine which architecture produces more universal features in the bottom network layers, $\mathcal{H}$. To do this, we pre-train $\mathcal{H}$ using SwAV on the same pre-train set for both networks. We then train the top layers $\mathcal{G}$ and output layer $\mathcal{F}$ on ImageNet-1K in an offline way on reconstructed features from $\mathcal{H}$. The compression locations in both models maintain the same spatial dimension of $14 \times 14$. Since ResNet18 has a larger channel dimension (256) than MobileNetV3-L (80), the compression error is relatively higher in ResNet18.

Table 4 depicts comparisons with augmentations and without augmentation. In both cases, MobileNetV3-L features yield better accuracy than ResNet18 features. Therefore, MobileNetV3-L produces more generalizable features than ResNet18. Furthermore, MobileNetV3-L requires $1.9\times$ fewer parameters than ResNet18. This comparison validates our rationale for choosing MobileNetv3-L.

Table 5: SIESTA's performance as a function of quantization layer based on ImageNet-1K without augmentation. Here, $\mathcal{S}$ is the tensor size ($r \times s \times d$), $\mathcal{P}$ is the number of trainable parameters in millions, $\mathcal{M}$ is the memory consumption in GB, $\mu$ is the average top-5 accuracy (%), $\alpha$ is the final top-5 accuracy (%), and $\mathcal{E}_{OPQ}$ indicates the OPQ quantization error. The ($\uparrow$) and ($\downarrow$) indicate high and low values to reflect optimum performance respectively.

| Layer | $\mathcal{S}$ | $\mathcal{P}(\downarrow)$ | $\mathcal{M}(\downarrow)$ | $\mu(\uparrow)$ | $\alpha(\uparrow)$ | $\mathcal{E}_{OPQ}(\downarrow)$ |
|---|---|---|---|---|---|---|
| 3 | $56^2 \times 24$ | 5.47 | 32.19 | **89.39** | **85.18** | **0.03** |
| 5 | $28^2 \times 40$ | 5.46 | 8.08 | 89.16 | 84.68 | 0.07 |
| 8 | $14^2 \times 80$ | 5.36 | 2.05 | 88.33 | 83.59 | 0.18 |
| 14 | $7^2 \times 160$ | **4.26** | **0.55** | 85.37 | 79.69 | 0.35 |

Table 6: Experimental results with augmentations based on ImageNet-1K. We constrain methods to 58.8M updates but allow them to use their own augmentation settings. The ($\uparrow$) and ($\downarrow$) indicate high and low values to reflect optimum performance respectively. $\mathcal{P}$ is the number of parameters in Millions, $\mu$ is the average top-5 accuracy, $\alpha$ is the final top-5 accuracy, $\mathcal{M}$ is the total memory in GB, and $\mathcal{U}$ is the total number of updates in Millions.

| Method | $\mathcal{P}(\downarrow)$ | $\mu(\uparrow)$ | $\alpha(\uparrow)$ | $\mathcal{M}(\downarrow)$ | $\mathcal{U}(\downarrow)$ |
|---|---|---|---|---|---|
| Offline | 5.48 | — | 90.74 | 192.87 | 753.33 |
| DER | 54.80 | 82.72 | 71.82 | 20.99 | 58.39 |
| ER | 5.48 | 82.19 | 71.22 | 19.59 | 58.78 |
| REMIND | 5.48 | 87.67 | 82.97 | 2.02 | 58.78 |
| **SIESTA** | **5.48** | **90.67** | **87.00** | **2.02** | **57.60** |

## F  MobileNet Quantization Layer

In Table 5, we examined SIESTA's utility (parameters and memory) and performance (accuracy and compression error) for various MobileNet quantization layers. We maintained identical OPQ configurations for this analysis. As we move quantization layer up towards input, SIESTA's accuracy increases due to increase in trainable parameters and OPQ quantization error decreases due to decrease in channel dimension. However, the increase in accuracy comes at a cost of increased memory. Since spatial dimension in upper layers increases, the memory requirement increases due to increased number of vectors corresponding to larger spatial dimension. For example, spatial dimension in layer 5 is $2\times$ larger than that in layer 8, hence memory required to store features in layer 5 becomes $4\times$ more than layer 8. The opposite is true when moving quantization layer down towards output. To balance accuracy and efficiency we select layer 8 for all our experiments in the main results. Layer 8 provides suitable tensor size to meet lower memory budget while maintaining optimum accuracy.

## G  Comparison with the Offline Learner when Augmentations are Used

In Table 6, we compare SIESTA with REMIND, ER, and DER under identical settings where all methods use MobileNetV3-L, SwAV pre-training, same number of updates, and augmentations. SIESTA outperforms DER by 15.18% (absolute) in final accuracy while using $10\times$ less memory and $10\times$ fewer parameters. SIESTA also outperforms ER by 15.78% (absolute) in final accuracy while using $9.7\times$ less memory. While SIESTA uses same number of parameters and same amount of memory as REMIND, it outperforms REMIND by 4.03% (absolute) in final accuracy. When comparing to the offline learner, SIESTA has the smallest gap (3.74% absolute) in terms of final accuracy.

Using McNemar's test to compare SIESTA's final top-5 accuracy in the iid and class incremental settings, there was no significant difference in the accuracy of the two systems ($P = 0.85$). Thus, SIESTA appears to be invariant to the order of the data when augmentations are used, which is consistent with our findings from Section 6.1 without augmentations. Using a McNemar's test, we observed that there was a significant difference between the offline model and SIESTA when augmentations were used for both orderings ($P < 0.001$). This is in contrast to our results without augmentations, where we found no significant difference. There are several routes that we believe are promising for closing this gap. These include using self-supervised learning methods that outperform SwAV, using additional capacity in $\mathcal{H}$, and developing additional augmentation techniques that could be used with the reconstructed tensors.

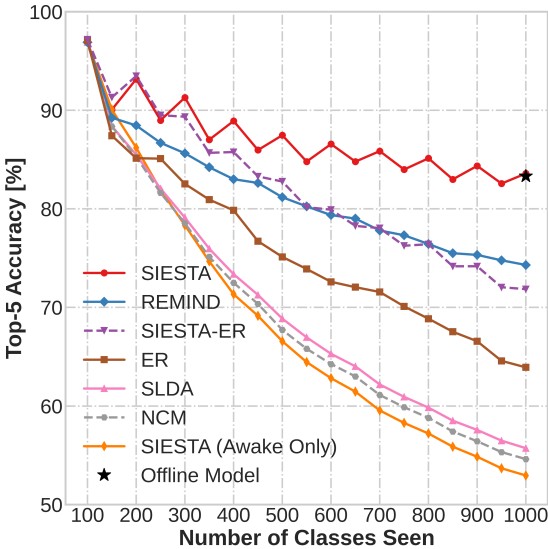

Figure 7: Learning curves for online continual learners without augmentations evaluated on ImageNet-1K in class-incremental setting. SIESTA (Awake Only) is an online variant of SIESTA that does not perform sleep. SIESTA-ER uses veridical rehearsal.

Table 7: Impact of buffer size on SIESTA's performance evaluated on ImageNet-1K without augmentation. The ($\uparrow$) and ($\downarrow$) indicate high and low values to reflect optimum performance respectively. $\mu$ and $\alpha$ denote average top-5 accuracy (%) and final top-5 accuracy (%), respectively. Buffer size is reported in GB.

| Buffer Size | Quantized Instances | $\mu(\uparrow)$ | $\alpha(\uparrow)$ |
|---|---|---|---|
| 0.75 | 479665 | 87.33 | 80.57 |
| 1.51 | 959665 | 88.30 | 83.30 |
| 2.01 | 1281167 | 88.33 | 83.59 |

## H   Online Learning Comparisons

In Figure 7, we show learning curves on ImageNet-1K comparing online learning methods with an awake-only variant of SIESTA, which does not perform sleep steps. SIESTA (Awake Only) is an online learning method that updates $\mathcal{F}$ as described in Section 4.1. From Figure 7, we can see that SIESTA (Awake Only) rivals popular online learning methods NCM and SLDA, whereas with sleep it vastly outperforms them. SIESTA exceeds REMIND, ER, SLDA, and NCM by 9.28%, 19.67%, 27.87%, and 28.98% (absolute) respectively in final accuracy.

We also compare a veridical rehearsal variant of SIESTA, SIESTA-ER with the widely compared veridical rehearsal baseline ER (Chaudhry et al., 2019). Both of them store same number of raw images in the buffer and use same number of network updates. As shown in Figure 7, SIESTA-ER outperforms ER by large margin in all increments. SIESTA-ER exceeds ER by 7.94% (absolute) in final accuracy while requiring $3\times$ less training time than ER. All models use the same SwAV pre-trained DNN.

## I   Impact of Buffer Size

In Table 7 we studied SIESTA's performance as we altered the size of the buffer. We kept the total number of updates during a sleep cycle constant, which is consistent with observations in the dataset pruning literature that even when using pruned datasets a comparable number of updates are necessary to get equivalent performance in a neural network (Sorscher et al., 2022). We see that SIESTA performs remarkably with smaller buffer size. The drop in final accuracy while changing buffer size from 2.01 GB to 0.75 GB is only 3.02% (absolute).

Table 8: Impact of batch size on SIESTA's performance evaluated on ImageNet-1K without augmentation. The ($\uparrow$) and ($\downarrow$) indicate high and low values to reflect optimum performance respectively. $\mu$ and $\alpha$ denote average top-5 accuracy (%) and final top-5 accuracy (%), respectively. Total training time T is reported in hours.

| Batch | T($\downarrow$) | $\mu$($\uparrow$) | $\alpha$($\uparrow$) |
|---|---|---|---|
| 32 | 3.80 | 88.36 | 83.53 |
| 64 | 2.39 | 88.33 | 83.59 |
| 128 | 2.20 | 88.38 | 83.72 |
| 256 | 1.97 | 88.34 | 83.61 |
| 512 | 1.89 | 88.29 | 83.50 |

Table 9: Comparison among different SSL pre-training methods. The top network ($\mathcal{G}$ and $\mathcal{F}$) is trained and evaluated on ImageNet-1K. Reported is best top-1 accuracy (%).

| Methods | Best Accuracy ($\uparrow$) |
|---|---|
| Supervised (Upper Bound) | 72.33 |
| Random Initialization | 67.02 |
| MIRA (Lee et al., 2022) | 67.73 |
| MIRA + Barlow Twins (Zbontar et al., 2021) | 63.69 |
| SEED (Fang et al., 2020) | 68.78 |
| **SwAV** (Caron et al., 2020) | **69.86** |
| SwAV + E-SSL (Dangovski et al., 2022) | 69.71 |
| SwAV + SEAL (Sarfi et al., 2023) | 69.66 |

## J  Batch Size and Training Time for SIESTA

In Table 8 we studied SIESTA's performance as we varied the batch size in offline training during sleep. We kept the total number of updates during a sleep cycle constant. We conducted the experiments on same hardware (NVIDIA RTX A5000 GPU). SIESTA shows robustness to varying batch size where it performs almost similarly for batch size ranges from 32 to 512. Increasing batch size provides a speed up, where a batch size of 512 requires $2\times$ less time than batch size 32 to finish ImageNet-1K training. Hence SIESTA becomes $4.3\times$ faster than REMIND on same hardware. This makes SIESTA the most performant and fastest online continual learning method to date.

## K  SSL Pre-training Analysis

In this section, we compare different SSL methods used to pre-train MobileNetV3-L. For pre-training, we follow default training procedure from respective SSL papers. After pre-training, we initialize MobileNetV3-L using SSL pre-trained weights. Next, we keep bottom part, $\mathcal{H}$ frozen and train top part ($\mathcal{G}$ and $\mathcal{F}$) on full ImageNet-1K in offline supervised manner for 50 epochs. For image augmentations, we apply random resized crop and random horizontal flip. The top part ($\mathcal{G}$ and $\mathcal{F}$) is trained on real features (without OPQ) extracted from $\mathcal{H}$. We also test naive *Random Initialization* where $\mathcal{H}$ is initialized using SwAV pre-trained weights and $\mathcal{G}$ and $\mathcal{F}$ are randomly initialized. In Table 9, we report best top-1 accuracy achieved by various methods any time during training.

From Table 9, we observe that all SSL methods show a gap in accuracy compared to supervised upper bound. This indicates that features in bottom network $\mathcal{H}$ learned by these SSL methods do not fully generalize to unseen classes. As a result, SIESTA struggles to match the performance of the offline learner when augmentations are used. Among compared SSL methods, SwAV has the lowest gap with supervised upper bound.

## L  Additional Datasets

Here we evaluate SIESTA, ER, and REMIND on a large-scale dataset, Places365-Standard (1.8 Million images). We also evaluate SIESTA, ER, and REMIND on the CUB-200 dataset. All experiments use class incremental learning setting without augmentation.

Table 10: Places365-Standard experimental results without augmentation. Symbols ($\uparrow$) and ($\downarrow$) indicate high and low values to reflect optimum results. $\mathcal{P}$ is the number of parameters in Millions, $\mu$ is the average top-5 accuracy, $\alpha$ is the final top-5 accuracy, $\mathcal{M}$ is the total memory in GB, and $\mathcal{U}$ is the total number of updates in Millions.

| Method | $\mathcal{P}(\downarrow)$ | $\mu(\uparrow)$ | $\alpha(\uparrow)$ | $\mathcal{M}(\downarrow)$ | $\mathcal{U}(\downarrow)$ |
|---|---|---|---|---|---|
| ER | 5.48 | 75.94 | 63.90 | 19.59 | 7.21 |
| REMIND | 5.48 | 81.48 | 73.52 | 2.02 | 7.21 |
| **SIESTA** | 5.48 | **87.60** | **81.07** | **2.02** | 7.21 |

Table 11: CUB-200 experimental results without augmentation. Symbols ($\uparrow$) and ($\downarrow$) indicate high and low values to reflect optimum results. $\mathcal{P}$ is the number of parameters in Millions, $\mu$ is the average top-5 accuracy, $\alpha$ is the final top-5 accuracy, $\mathcal{M}$ is the total memory in MB, and $\mathcal{U}$ is the total number of updates in Thousands. For $\mu$ and $\alpha$, we report the mean $\pm$ standard deviation across 6 runs.

| Method | $\mathcal{P}(\downarrow)$ | $\mu(\uparrow)$ | $\alpha(\uparrow)$ | $\mathcal{M}(\downarrow)$ | $\mathcal{U}(\downarrow)$ |
|---|---|---|---|---|---|
| ER | 5.48 | $76.30 \pm 0.73$ | $74.62 \pm 1.41$ | 902.26 | 41.96 |
| REMIND | 5.48 | $74.00 \pm 1.44$ | $70.76 \pm 1.94$ | 31.40 | 41.96 |
| **SIESTA** | 5.48 | $\mathbf{86.51 \pm 0.18}$ | $\mathbf{83.52 \pm 0.28}$ | **31.40** | **38.40** |

For Places365-Standard dataset, we first initialize MobileNetV3-L with SwAV weights pre-trained on ImageNet base initialization subset i.e., 100 classes. After base initialization, each model learns 365 classses in a total 5 steps where each step consists of 73 classes. Each model uses the same number of total updates, which corresponds to SIESTA using $m = 1.44M$ updates per sleep cycle whereas ER and REMIND use 3 rehearsal samples per mini-batch. The total number of updates for end-to-end training is bounded by 7.21M. For Places365-Standard, we apply same memory constraints as ImageNet experiments. SIESTA uses same settings as described in Section 5. REMIND uses a fixed LR of 0.01 and the LR scheduler is omitted due to degraded performance. Other implementation details follow the original REMIND paper. ER uses the SGD optimizer with a fixed LR of 0.01. Other implementation details follow our ImageNet experiments without augmentation from Sec. C.6.

For CUB-200 dataset, we first initialize MobileNetV3-L with SwAV weights pre-trained on ImageNet base initialization subset i.e., 100 classes. After base initialization, each model learns 200 classes in 4 steps where each step consists of 50 classes. Each model uses the same number of total updates, which corresponds to SIESTA using $m = 9600$ updates per sleep cycle whereas ER and REMIND use 6 rehearsal samples per mini-batch. The number of updates for end-to-end training is bounded by 42K. Since CUB-200 is a small dataset with 5994 training images, we store all samples in the memory buffer. SIESTA uses an initial LR of 0.3 for batch size 32. Other implementation details adhere to Section 5. REMIND uses the same implementation as used in the REMIND paper except for the LR. We set the initial LR and the final LR to 0.03 and 0.0003, respectively. ER uses the SGD optimizer with a fixed LR of 0.03.

We summarize experimental results in Table 10 and Table 11. For both datasets, SIESTA achieves the highest accuracy and outperforms compared methods by a large margin. In particular, SIESTA outperforms ER and REMIND by 17.17% and 7.55% (absolute) in final accuracy on the Places365-Standard dataset. We also observe that SIESTA is 4.4$\times$ faster to train on the large-scale Places365-Standard dataset compared to REMIND using the same hardware. SIESTA also outperforms ER and REMIND by 8.9% and 12.76% (absolute) respectively in final accuracy on the CUB-200 dataset.

