# OpenReview forum: "SIESTA: Efficient Online Continual Learning with Sleep"
_TMLR — Accepted by TMLR_

### Review · Reviewer_KcW5 · 2023-09-07

**Summary Of Contributions:**

Authors present a computationally efficient (less parameters, memory, compute) continual learning algorithm which manages to achieve SotA on class-incremental Imagenet-1K.

**Audience:**

Yes

**Claims And Evidence:**

No

**Requested Changes:**

- The significance of the results should be better supported. It would be much more convincing with comparisons to more recent work (there is only one paper after 2021, DyTox), ideally 2-3 (critical for recommendation for acceptance)
- There are some over-claims (critical for recommendation for acceptance)
  - in the abstract -> "online" CL method, but it is not purely online, as it requires offline phases + offline pre-training.
  - "solves catastrophic forgetting" pg 3. The results do not support this claim. It would be good to qualify the statement so that it is actually supported by the results, ie . "solves catastrophic forgetting given XYZ assumptions" - critical for recommendation for acceptance
- Acknowledging the limitations more would make this paper stronger, for eg. to understand when it is appropriate to be used in more realistic settings (not critical for recommendation for acceptance)

**Strengths And Weaknesses:**

Strengths
- Imagenet-1K results are impressive
- online update rule is computationally light-weight (no gradient updates)
- computationally scales better with increasing number of incremental batches (larger datasets) than prior-arts
- empirical study of optimal architecture, suggesting CL practitioners should move away from using ResNet-18
- writing and structure is coherent

Weaknesses
- the approach is an amalgamation of several existing strategies which have been studied in CL. In this circumstance, isolating the contribution and providing insights about each of the "moving parts" would make the paper more interesting
  - use of pre-trained feature extractors (including via SSL)
  - tensor quantization strategy:  How much of the reduction in memory/compute cost is due to this?
  - wake-sleep algorithms have been explored in CL, in addition to the references given in the paper (see [1],[2] for a couple of notable examples). The idea of "awake" and "sleep" phase is very general, what new insight does this work bring?
  -  the online update rule (eq. 3) is similar to online prototype learning and the same is presented in another CL work (eq. 1) [3] . Were other update rules explored ? Why use this one in particular?

- Authors compare performance between their CNN-based architecture with pre-training and transformer-based approach trained from scratch (DyTox). These factors would intuitively result in DyTox requiring more updates to achieve similar performance. Overall, the comparisons with recent work leave a bit to be desired.
- SIESTA performs worse than online-CL methods (SLDA, NCM) in awake-only version (Pure online), suggesting offline phase is necessary for this approach to be useful. Is there any insight as to why it is less competitive in the online-only setting?

[1] Lee, S., Ha, J., Zhang, D., & Kim, G. (2020). A neural dirichlet process mixture model for task-free continual learning. arXiv preprint arXiv:2001.00689.

[2] Li, S., Du, Y., van de Ven, G. &amp; Mordatch, I.. (2022). Energy-Based Models for Continual Learning. <i>Proceedings of The 1st Conference on Lifelong Learning Agents</i>, in <i>Proceedings of Machine Learning Research</i> 199:1-22 Available from https://proceedings.mlr.press/v199/li22a.html.

[3] He, Jiangpeng, and Fengqing Zhu. "Exemplar-free online continual learning." 2022 IEEE International Conference on Image Processing (ICIP). IEEE, 2022.

---

> ### Author Response · Authors · 2023-09-25
> **We have included new experiments and revised our paper based on reviewer's comments**
>
> We thank the reviewer for the constructive comments! We have edited our paper to reflect our responses and address the reviewer's concerns. If additional details or clarifications are needed, we will be happy to provide them.
>
> W1: We conducted additional ablation experiments where we analyzed each component of SIESTA separately (Sec. 6.5). In all cases, SIESTA showed efficacy and robustness.
>
> W1.1: Following prior CL works, we defined our problem setting with a base initialization phase where the model acquires base knowledge based on a pre-train dataset (10% of ImageNet-1K). Many CL works perform this base initialization step to obtain a pre-trained feature extractor. For fair comparison, we allowed all compared methods e.g., ER, DER, and REMIND to use the same pre-trained backbone as used by SIESTA.
>
> W1.2: For memory efficiency, we used tensor quantization. Compared to storing raw images, storing quantized samples leads to 96x reduction in memory footprint. We also examined the impact of memory on SIESTA’s performance in Appendix I.
> On the other hand, tensor quantization does not impact compute cost which is defined by total number of samples used to compute SGD updates regardless of the nature of samples (raw or quantized). During training, if a model uses N raw samples (RGB images) whereas another model uses N quantized samples, then they both have the same compute cost (U=N).
>
> W1.3: Yes, we mentioned in the paper that wake and sleep phases were also explored by other works. However, there is no prior work that proposes a wake-sleep framework with ability to scale to a large-scale dataset like ImageNet-1K and achieve identical performance to the offline model while using far less compute and training time. In contrast, SIESTA demonstrates that the wake-sleep framework enables efficient updates of DNN using far less compute (11.53M updates) and training time (1.9 hours) than SOTA methods.
> SIESTA’s wake-sleep framework has novel contributions toward efficient CL especially on-device learning. For example, SIESTA allows mobile devices to quickly use novel observations from users and their environments and then consolidate that learning during a scheduled downtime when devices are being charged.
>
> W1.4: The online update of class means in mentioned paper is similar to ours however there are significant differences between two methods. They perform the online update of class means during training and use the nearest class mean classifier (NCM) during inference where NCM uses class means to make predictions. In contrast, SIESTA performs the online update of class means and also updates corresponding weights in the final layer (F) using class means. During inference, SIESTA uses cosine similarity scores (equation 1 and 2) to make predictions where higher cosine similarity scores with respect to class means lead to correct predictions.
> We also explored NCM and SLDA update rules besides our proposed rule, however they both performed sub-optimally.
> We used the proposed network update rule to align the cosine softmax loss function used during sleep with the online updates during the wake period.
>
> W2: Following prior work, we defined our problem setting with a base initialization phase which happens only once in a model’s lifetime before CL begins. Hence we excluded base initialization (first batch for other methods) from compute and memory overhead comparisons and only considered the CL phase for these comparisons.
>
> W3: Since SIESTA awake-only depends on representations learned on base initialization dataset, those representations limit its ability to some extent (Appendix K). When representations were updated during the sleep phase, SIESTA achieved the highest performance during both sleep and wake phases (Fig. 7).
>
> RC1: Now we have included additional recent works such as MEMO [1] and FOSTER [2] in Table 2 where SIESTA outperformed them in all criteria. In particular, SIESTA (Aug) outperformed MEMO and FOSTER by 6.11% and 15.15% (absolute) respectively in final accuracy on the ImageNet-1K dataset.
>
> [1] Da-Wei Zho et al., A model or 603 exemplars: Towards memory-efficient class-incremental learning, In ICLR 2023.
>
> [2] Fu-Yun Wang et al, Foster: Feature boosting and compression for class-incremental learning, In ECCV 2022.
>
> RC2.1: We proposed an “online updates with offline consolidation” paradigm to target real-world applications especially on-device learning. In this paradigm, SIESTA performs online CL during wake phase and performs offline learning during sleep phase. We have omitted “online” in the abstract.
>
> RC2.2: Now, we have modified that statement based on results in Table 1 where SIESTA matched offline accuracy. We have stated that “It suffers from zero catastrophic forgetting in the augmentation-free setting”.
>
> RC3: We discussed limitations of current SIESTA in Sec. 6.1 and Sec. 7. We mentioned that SIESTA could not achieve identical performance to the offline model in experiments with augmentations.

---

### Review · Reviewer_e6v6 · 2023-09-14

**Summary Of Contributions:**

The paper addresses continual learning, an important problem in deep learning. The goal is to avoid catastrophic forgetting when training neural networks. Unlike i.i.d. training, in continual learning settings, the data is passed into the network in a continuous stream and one often cannot revisit the old data unless they have extra memory. The core idea of the paper is to introduce a wake/sleep mechanism with memory replay. The feature representation of the input data is quantized and stored in the memory. During the sleep phase, the features are restored and replayed to the neural network (in this case, the neural net is trained in an i.i.d. manner). The experiments demonstrate the power of this method, which is impressive, achieving the same level of accuracy compared to purely offline training (i.i.d.). The paper also shows its strength in memory and computational cost. Overall, I think this is a nice paper.

**Audience:**

Yes

**Broader Impact Concerns:**

What if some data is not clean or incorrect labels are provided by attackers, how would this method react? The fact that the method utilizes a memory and replay the labels, basically assumes the data is clean. However, in real world, the data/labels can be changed which may leads to catastrophic failure of the model. It'll be good to discuss around this questions and potentially provide some results on this.

**Claims And Evidence:**

Yes

**Requested Changes:**

Please consider improving the paper and provide responses according to the weaknesses.

In addition to the above, few more questions to address:

1. Why does the wake phase need to finetune the last layer F? Either way, in the sleep phase, you will finetune G and F. I don't see a strong need for doing this in the wake phase. Is it more for the purpose of increasing the average performance during the whole training process?
2. I believe by finetuning the last layer F, you still have forgetting effects. Do you see this as a problem?
3. Effectively the method divides the network into three pieces and uses three different learning rates. I find picking the right learning rates may be critical here. How do you choose them? Do you have data to support their selection is not sensitive?

**Strengths And Weaknesses:**

Strengths:
1. The paper addresses an important problem and the proposed method is technically sound.
2. The paper's idea is very interesting, which somehow relates to how humans and animals learn during their sleep cycles.
3. The experimental validation shows the power of the method, beating state-of-the-art methods and surprisingly achieving the same level of accuracy compared to i.i.d. offline training (which is impressive).
4. The paper also provides necessary ablation studies which demonstrate the value of sleep mechanisms.

Weaknesses:
1. The paper seems to only evaluate the method on one dataset. readers would be curious about the performance on other standard continual learning datasets. I understand that traditional continual learning datasets are very small but would be nice to test it on one of them. Could be interesting to consider the setting in Hu et al. One Pass ImageNet (https://arxiv.org/pdf/2111.01956.pdf) and provide a discussion on how the proposed method works in that large-scale setting.
2. The community is shifting to transformers while the paper is focusing on ResNets. Would be great to consider how the proposed method works for transformer-based neural architectures.
3. The hyperparameter of the number of layers for H seems pretty hard to determine. Can you explain more on how people can choose this value effectively?

---

> ### Author Response · Authors · 2023-09-25
> **We have included new experiments and revised our paper based on reviewer's comments**
>
> We thank the reviewer for the constructive comments! We have edited our paper to reflect our responses and address the reviewer's concerns. If additional details or clarifications are needed, we will be happy to provide them.
>
> W1: Besides ImageNet-1K, we also evaluated SIESTA on a long-tailed dataset, Places365-LT in Appendix D where SIESTA outperformed REMIND (Table 3). Recently, we have conducted experiments on two additional datasets e.g., Places365-Standard and CUB-200 (see Appendix L).  We find that SIESTA outperformed ER and REMIND by 17.17% and 7.55% (absolute) respectively in final accuracy on the Places365-Standard dataset (Table 10). SIESTA also outperformed ER and REMIND by 8.9% and 12.76% (absolute) respectively in final accuracy on the CUB-200 dataset (Table 11). Overall, SIESTA showed performance superior to other methods on four datasets.
>
> SIESTA does not perform multiple passes over the ImageNet dataset which is similar in spirit to One Pass ImageNet (OPIN) paper. Also, we reported memory and compute overheads of CL methods and the offline model in all comparisons (Table 1 and 6). So a relative compute and memory comparison with respect to offline model can be made similar to OPIN where SIESTA still outperforms other methods in all criteria.
>
> W2: We did not focus on ResNets. Since our focus is efficient on-device learning, we considered mobile platform friendly architecture, MobileNetV3-L. We found that MobileNetV3-L outperformed ResNet18 on ImageNet-1K dataset (Table 4).
> SIESTA is a general framework and can be implemented with other classes of architecture as long as tensor embeddings can be quantized. We discussed this in section 7. For future work, SIESTA can be extended with transformer models for efficient CL.
>
> W3: We conducted extensive experiments to examine the number of layers for H in Appendix F. We measured how the final performance and memory usage changed as we varied the number of frozen layers. To balance accuracy and memory efficiency we selected layer 8 which corresponds to only 2.19% of the network parameters.
> One can select the number of layers for H based on 2 criteria: 1) memory footprint (size of embedding tensors) and 2) performance (trainable layers/ parameters). For a given dataset and network architecture, similar experiments can be done to make an informed decision about the number of layers for H and find the best trade-off.
>
> Q1: SIESTA’s wake-sleep framework fits into embedded applications for example, SIESTA allows mobile devices to quickly use novel observations from users and their environments and then consolidate that learning during a scheduled downtime when devices are being charged.
> For these applications, during its wake phase, SIESTA learns new information in an online manner before the sleep phase begins where it consolidates new knowledge using offline learning. Online learning happens via updating final layer weights using the proposed network update rule. Without online learning, SIESTA does not have knowledge on new data and cannot perform any predictions on newly observed data before sleep occurs. We want SIESTA to learn new information and use its new knowledge while awake so that SIESTA can interact with users or environments during the wake phase where its users can personalize it and use it immediately.
>
> Q2: We observe that updating the last layer F causes humble forgetting and hence we do not see this as a problem. Because the penultimate embedding space remains sufficiently discriminative due to the cosine softmax loss objective. Consequently, class means do not have much inter-class interference and SIESTA produces higher cosine similarity for the correct label since samples of the same class have higher cosine similarity with corresponding class mean.
> If we observe differences in accuracy on old data (tasks) due to online updates of the final layer F, we find that average forgetting is only 5.5% across 9 incremental steps in class incremental learning on ImageNet-1K.
>
> Q3: Although SIESTA splits the DNN into three parts, it does not require 3 different learning rates or settings. It uses one universal setting i.e., same network configuration, same optimizer, same learning rate scheduler, and same learning rate for all learning phases. We have mentioned this in section 5.
> SIESTA does not require any parametric updates during its wake phase which is free from backprop and hyperparameter tuning.
> During the sleep phase, SIESTA trains DNN with parametric updates, however its hyperparameters are tuned once and kept fixed for all sleep cycles. All experiments including ImageNet-1K, Places365-Standard, Places-LT, and CUB-200 are based on the same setting as described in Sec. 5.
>
> Border Impact Concerns: We recognize that studying label noise is critical for real-world applications as obtaining 100% clean data might be difficult. Now, we have included a discussion about this in Sec. 7 (last paragraph). We thank you for your insightful feedback.

---

### Review · Reviewer_GKA2 · 2023-09-18

**Summary Of Contributions:**

# Summary:

The paper titled "SIESTA: Efficient Online Continual Learning with Sleep" introduces SIESTA, a novel online continual learning (CL) method designed to efficiently update deep neural networks with ever-growing data streams. SIESTA operates on a wake/sleep framework and focuses on computational efficiency for both on-device learning and large production-level DNNs. It innovatively combines rapid online updates during the wake phase and memory consolidation during the sleep phase, using quantized latent rehearsal. The paper presents results demonstrating that SIESTA outperforms existing CL methods, achieves identical performance to an offline learner in some scenarios, and excels in terms of computational and memory efficiency.

The paper's significance lies in its contribution to addressing the challenges of continual learning in real-world applications, specifically for the edge devices. SIESTA's focus on computational efficiency and its ability to match or exceed the performance of offline learners make it a valuable addition to the field of continual learning, potentially opening up new possibilities for on-device learning and efficient model updates. The wake/sleep framework, the use of quantized latent rehearsal, and the achievement of performance parity with offline learners in certain scenarios represent novel contributions to the field of continual learning.

**Audience:**

Yes

**Broader Impact Concerns:**

# Broader Implications:

The paper's focus on efficient continual learning has broader implications for various industries, including on-device learning for mobile devices, robotics, augmented reality, and more. Efficient CL methods like SIESTA can reduce the computational and energy costs of model updates, making them more feasible for real-world applications with ever-growing data. This could lead to more adaptive and efficient AI systems in the future.

**Claims And Evidence:**

Yes

**Requested Changes:**

# Suggestions for Improvement:

1. Consider experimenting with a wider range of datasets and application scenarios to demonstrate SIESTA's versatility.

2. Explore the sensitivity of SIESTA to hyperparameters more comprehensively to understand its generalization to different settings.

**Strengths And Weaknesses:**

# Strengths:

1. **Efficiency Achievement:** SIESTA stands out for its remarkable computational and memory efficiency compared to existing CL methods. It outperforms these methods while requiring significantly fewer network updates, parameters, memory, and computational resources. SIESTA's use of quantized latent rehearsal for memory efficiency is a notable innovation, enabling it to store more samples with limited memory.

2. **Real-World Applicability:** The paper emphasizes the practical applicability of CL for real-world scenarios by addressing the challenges of on-device learning and the need for continual updates in large production systems.

3. **Innovative Framework:** SIESTA introduces the wake/sleep framework for CL, inspired by the way humans and animals learn, where online learning is consolidated during offline sleep. This approach aligns with the demands of many real-world scenarios, where the learning algorithm can update itself when it is left idle.

# Weaknesses:

1. **Limited Dataset:** The paper primarily focuses on experiments with ImageNet-1K and Places, which, while significant, might not fully represent the diversity of real-world applications. More diverse datasets and application scenarios could strengthen the paper's claims.

2. **Complexity and Parameter Tuning:** The SIESTA algorithm, while highly effective, comes with increased complexity due to its wake-sleep framework and multiple phases of operation. This complexity requires further parameter tuning to adapt it to different datasets and hardware configurations, there incurring a computational overhead.

3. **Augmentation Dependency:** SIESTA's augmentation experiments show significant performance gains, but they also highlight its sensitivity to data augmentation techniques. Depending on the specific dataset and augmentation strategies used, SIESTA's performance may vary, and it may not always outperform other methods. This dependence on data augmentation could limit its robustness in scenarios where augmentation is challenging or not feasible.

---

> ### Author Response · Authors · 2023-09-25
> **We have included new experiments and revised our paper based on reviewer's comments**
>
> We thank the reviewer for the constructive comments! We have edited our paper to reflect our responses and address the reviewer's concerns. If additional details or clarifications are needed, we will be happy to provide them.
>
> W1 (Limited Dataset): Recently, we have conducted experiments on two additional datasets e.g., Places365-Standard and CUB-200 (see Appendix L).  We find that SIESTA outperformed ER and REMIND by 17.17% and 7.55% (absolute) respectively in final accuracy on the Places365-Standard dataset (Table 10). SIESTA also outperformed ER and REMIND by 8.9% and 12.76% (absolute) respectively in final accuracy on the CUB-200 dataset (Table 11). We also observe that SIESTA is 4.4x faster to train on the large-scale Places365-Standard dataset (1.8M training images) compared to REMIND using the same hardware. We have revised our paper with these new results.
>
> Overall, SIESTA showed performance superior to other methods on four datasets including two large-scale datasets (ImageNet-1K and Places365-Standard), one long-tailed dataset (Places365-LT), and one small dataset (CUB-200).
> We also evaluated SIESTA using various data orderings such as class incremental learning and continual IID to simulate various application scenarios where SIESTA maintained identical performance and compared methods performed differently (see Sec. 6.1). This indicates that SIESTA generalizes to various CL scenarios while maintaining competitive performance compared to the offline model.
>
> W2 (Complexity and Parameter Tuning): Although SIESTA is based on wake-sleep framework, it uses the same universal setting i.e., same network configuration and same hyperparameters for all learning phases. We have now mentioned this in section 5.
> Its online learning of the output layer during the wake phase is free from any parametric updates and hyperparameters. Online learning is based on a data-driven network update rule where SIESTA updates its output layer weights using class means. Therefore, online learning appears to be a general approach for any datasets.
>
> On the other hand, offline learning during the sleep phase includes parametric updates; however, its hyperparameters are tuned once and kept fixed for all sleep phases. All experiments including ImageNet-1K, Places365-Standard, Places365-LT, and CUB-200 as well as class incremental learning and continual IID are based on the same setting i.e., same hyperparameters, same optimizer, same learning rate scheduler, and same network configuration.
> For a different dataset, one can tune hyperparameters once and use the same setting for all learning phases in SIESTA.
>
> W3 (Augmentation Dependency): Augmentation benefits SIESTA similarly as other CL methods as well as the offline model. If we compare each method’s performance without augmentation (Table 1) and with augmentation (Table 6), we observe that SIESTA, REMIND, ER, DER, and offline model achieved performance gain of 3.41%, 8.66%, 7.30%, 1.67%, and 7.43% (absolute) respectively in final accuracy on ImageNet-1K due to augmentation. Here SIESTA achieved moderate performance gain compared to REMIND that achieved the highest performance gain.
>
> As discussed in Appendix B, many CL methods apply various image augmentations and these differ across these systems. For example, BiC and DER use random crops and horizontal flips, while DyTox uses Rand-Augment. To compare CL methods without impact of augmentation strategies, we omitted augmentations in our main experiments in Sec. 6.1. This allowed us to compare CL algorithms in a fair way, where we also used the same DNN architecture. In these comparisons, SIESTA outperformed other methods.
> For a different dataset, one can use the same augmentation strategies i.e., mixup and cutmix as used in ImageNet experiments for SIESTA algorithm. We believe that SIESTA will rival the offline model without augmentation and show similar relative performance gain as offline model and other CL methods when augmentations are used.
>
> S1: We thank you for the valuable suggestions. Recently, we have conducted experiments on two additional datasets e.g., Places365-Standard and CUB-200 (see Appendix L).  We have revised our paper with these new results.
> We evaluated SIESTA in both class incremental learning (CIL) and continual IID settings to demonstrate its versatility and applicability for various data orderings. We found that unlike compared methods, SIESTA maintains identical performance on both settings whereas other methods, e.g., REMIND and ER showed drops in performance when switching from continual IID to CIL (see Sec. 6.1).
>
> S2: SIESTA uses the same network configuration and hyperparameters for all experiments including ImageNet-1K, Places365-Standard, Places-LT, and CUB-200 as well as class-incremental learning and continual IID settings, demonstrating that SIESTA is not sensitive to hyperparameters. Hyperparameters can be tuned once using a small subset of data and kept fixed for all learning phases.

---

### Author Response · Authors · 2023-10-31
**Uploaded a camera ready version**

Dear Reviewers and Editorial Team,

Thank you so much for the insightful discussion and time spent reviewing our work. We are grateful for the opportunity to publish our work at TMLR and have uploaded a camera ready version.

Kind regards, the authors.

---

### Decision · Action_Editor_LAGH · 2023-10-27

**Recommendation:** Accept as is

**Comment:**

The reviewers seemed to find this submission to be well-suited for TMLR but also outlined several weaknesses in the original submission. The authors were able to address the relevant ones given TMLR's acceptance criteria.

In their final assessment, the reviewers all found the authors' response to be adequate and the latest version of the paper (notably the consistent results across the new benchmarks) to have addressed their concerns.

The reviewer also notes that some of its results are noteworthy and that this work has the potential to spark several additional works.

I am pleased to recommend that this work be accepted as is. Congratulations!

**Audience:**

This paper proposed a new continual learning method aimed at more practical settings. I do not doubt that this is of interest to some members of the TMLR audience and notably will be interesting for the continual-learning researchers.

**Claims And Evidence:**

In the current version of the work, reviewers all agree that all claims are correctly supported.